# MTBench: A Multimodal Time Series Benchmark for Temporal Reasoning and Question Answering

## Abstract

Understanding the relationship between textual data and time-series evolution is a critical yet under-explored challenge in applied data science. While multimodal learning has gained traction, existing time-series benchmarks provide limited support for evaluating cross-modal reasoning and complex question answering, both essential for capturing interactions between narrative information and temporal patterns. To bridge this gap, we introduce *Multimodal Time Series **Bench**mark* (**MTBench**), a large-scale benchmark designed to evaluate large language models (LLMs) on the joint reasoning over time-series and text, exemplified through financial and weather domains. MTBench consists of paired time-series and textual data, including financial analysis with aligned stock price movements and weather reports matched to historical temperature records. Unlike existing benchmarks focused on isolated modalities, MTBench offers a comprehensive testbed for language models to jointly reason over structured numerical trends and unstructured textual narratives. MTBench supports diverse tasks that require a deep understanding of both text and time-series data, including forecasting, semantic and technical trend analysis, and news-driven question answering (QA). These tasks assess the model's ability to capture temporal dependencies, extract key insights from text, and integrate cross-modal information. We benchmark state-of-the-art LLMs on MTBench, providing a systematic analysis of their effectiveness in capturing the causal relationships between textual narratives and temporal patterns. Our findings reveal significant challenges in current models, including difficulty with long-term dependencies, limited causal interpretation in financial and weather dynamics, and insufficient multimodal fusion. MTBench establishes a foundation for advancing multimodal time-series research and for developing the next generation of multimodal models capable of reasoning across narrative and time series data.

## 1 Introduction

The integration of time-series and textual data is critical for understanding complex real-world phenomena, where numerical trends and contextual narratives jointly facilitate comprehensive analysis and decision-making. While Large Language Models (LLMs) have shown remarkable progress in natural language processing, their ability to reason across both time series and text remains under-explored. Domains like finance and weather forecasting exemplify this need, where numerical data (*e.g.,* stock prices, temperature readings) is inherently intertwined with textual information (*e.g.,* financial reports, weather summaries) (Kurisinkel et al., 2024; Dueben & Bauer, 2021; Publications, 2020; 2023). Despite the progress in LLMs, there remains a gap in evaluating their capability to jointly interpret and reason over such multimodal datasets.

Previous multimodal timeseries–text datasets mainly focus on predictive tasks, such as forecasting, or treating text as auxiliary metadata to boost accuracy (Pan et al., 2024; Cao et al., 2023; Jin et al., 2023; Liu et al., 2025b; Wang et al., 2024a; 2025; Huber et al., 2023; Dong et al., 2024). This narrow scope overlooks reasoning-centric challenges such as analytical thinking, causal inference, and cross-modal insight, which are essential for real-world applications ranging from financial risk assessment to climate analysis (Xie et al., 2023; 2024; Hirano, 2024; Fons et al., 2024; Islam et al., 2023; Cai et al., 2024). Moreover, existing benchmarks rarely capture the intricate semantic alignment required

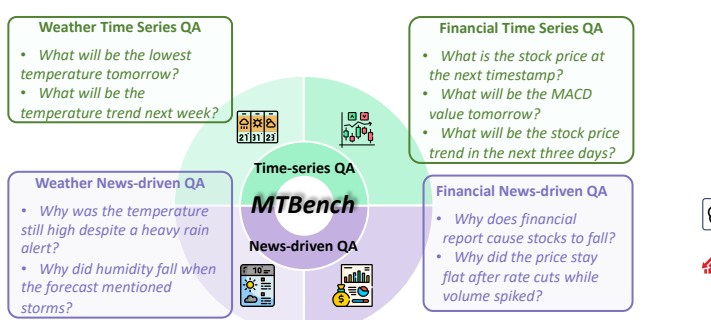
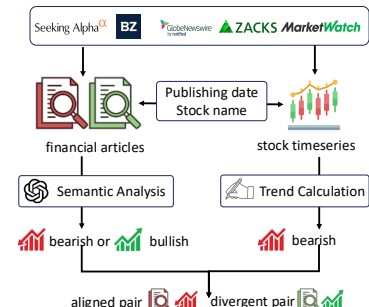

Figure 1: Overview of MTBench tasks across Time-series QA and News-driven QA in finance and weather domains.

Figure 2: Pipeline of Finance data collection and semantic alignment

between time-series signals and textual narratives. For example, a financial report might convey optimism, while the corresponding stock trend shows a downturn, revealing an inconsistency that exposes contradictions models must reconcile to ensure reliability. Without such considerations, models risk excelling at surface-level pattern matching yet failing at deeper reasoning tasks that demand causal interpretation and contextual understanding.

To overcome the aforementioned limitations, we introduce MTBench, a benchmark designed to evaluate LLMs on multi-task reasoning across time-series and text. As shown in Figure 1, unlike prior datasets that treat modalities independently or largely restrict themselves to time-series QA, such as forecasting or trend prediction, MTBench emphasizes meaningful cross-modal interactions by aligning time-series data with semantically relevant textual information. Moreover, MTBench extends beyond simple forecasting to include semantic trend analysis, technical indicator prediction, and news-driven question answering, which require reconciling narrative and numerical information.

In the financial domain, MTBench integrates high-resolution stock price series with contemporaneous financial news and reports, pairing them by both publication timing and semantic relevance (Figure 2), which ensures that textual evidence reflects causal and temporal relationships with market movements. For the weather domain, MTBench synchronizes temperature, humidity, and other meteorological variables from 50 U.S. weather stations with corresponding textual forecasts and reports. This pairing ensures that natural language descriptions faithfully track evolving climate patterns, thereby testing models on their ability to reconcile quantitative signals with qualitative narratives.

We evaluate state-of-the-art LLMs on MTBench across diverse reasoning tasks. Our findings show that LLMs consistently struggle with tasks requiring nuanced temporal understanding and effective integration of textual narrative information. Notably, incorporating relevant textual information generally improves LLMs' performance on time-series tasks, while time-series data also plays a key role in improving accuracy in news-driven QA. These results underscore the significance of multimodal context for advanced temporal reasoning and systems that can effectively integrate heterogeneous information sources.

Our contributions are threefold: (1) We introduce a multimodal time-series benchmark that uniquely extends beyond prediction to include reasoning and, for the first time, question answering tasks. (2) MTBench captures complex interactions between temporal signals and textual narratives, providing a systematic testbed for cross-modal understanding. (3) We design a flexible framework that supports adjustable temporal windows, task complexity, and input granularity, enabling diverse experimental settings and more robust analysis.

## 2 RELATED WORK

**LLMs for Time-Series Tasks**. Recent studies have explored using Large Language Models (LLMs) for time-series tasks such as forecasting, anomaly detection, and financial modeling. Techniques include aligning embeddings with time-series signals (Pan et al., 2024; Cao et al., 2023; Chen et al., 2025; Chang et al., 2025), reprogramming inputs with textual prompts (Jin et al., 2023; Liu et al., 2025b; Wang et al., 2024a), and integrating contextual information like stock metadata or events (Yu

et al., 2023; Wang et al., 2025). These works highlight the need for foundation models tailored to time-series data and comprehensive benchmarks for evaluating multimodal data scenarios.

**Time-series Benchmarks**. While many time-series LLM studies are evaluated primarily on cases with time-series–only inputs(Jin et al., 2023; Wang et al., 2024a; Goswami et al., 2024; Tan et al., 2025; Liu et al., 2025b; Ansari et al., 2024), there has been growing interest in constructing paired text–time-series datasets to better benchmark models' ability to capture temporal behaviors (Karger et al., 2024; Cai et al., 2024; Liu et al., 2025a). For example, Time-MMD (Liu et al., 2025a) combines textual and time-series data across multiple domains, but its time-series resolution is very limited, with most domains containing fewer than 1,000 sample points. ForecastBench (Karger et al., 2024) introduces a dataset designed to generate diverse and meaningful forecasting questions, yet the benchmark is primarily tailored to discrete event forecasting. TimeseriesExam (Cai et al., 2024) constructs multiple-choice exam-style questions with controlled difficulty levels to measure model performance; however, the questions are abstractly designed and do not explicitly account for application-driven contexts such as financial information or weather patterns. Other works have developed benchmarks tailored to specific domains, including medicine (Chan et al., 2024), environmental monitoring (Lin et al., 2024), energy systems (Alnegheimish et al., 2024), and transportation (Lan et al., 2024). In contrast, MTBench emphasizes real-world evaluation by supporting multiple tasks grounded in domain-specific usage, thereby providing a more realistic testbed for multimodal reasoning. A comparison between MTBench and other relevant time series benchmarks is shown in Table 1.

**Financial News Benchmarks**. Most existing financial benchmarks are restricted to a single modality, either text or time series, limiting their ability to assess multimodal reasoning in financial LLMs (Islam et al., 2023; Wang et al., 2024b; Liu et al., 2024b; Islam et al., 2024). More recent efforts have attempted to align time-series with textual data (Xie et al., 2023; 2024), but these datasets rely heavily on social media content, which lacks the structured semantics and reliability of formal financial reporting. FNSPID (Dong et al., 2024) provides a more grounded alignment between stock prices and financial news, but its scope remains confined to price prediction tasks. In contrast, our benchmark extends beyond prediction to encompass a broader range of tasks, such as financial indicator forecasting and multimodal question answering.

**Weather Benchmarks**. Most existing weather benchmarks have been developed for numerical weather prediction (Rasp et al., 2020; 2023; Kaggle, 2024; Menne et al., 2012; for Environmental Information, 2022). More recent work has explored integrating structured meteorological time-series with textual information for forecasting tasks (Huber et al., 2023). However, these benchmarks still fall short in supporting rigorous evaluation of LLMs on multimodal understanding due to the lack of high-quality, aligned textual data. To address this gap, MTBench aligns meteorological variables with textual descriptions by synthesizing news-style narratives from severe weather event reports, expanding both the temporal coverage and spatial diversity of existing resources.

Table 1: A Comparison between MTBench and relevant benchmarks. 📈 and 📄 indicate time-series and text modalities in queries.

| Benchmark | Domain | Query Format | Forecasting | Trend/Indicator | Complex QA | Narrative Context |
|---|---|---|---|---|---|---|
| Time-MMD (Liu et al., 2025a) | Generic | 📈 | ✓ | ✗ | ✗ | ✓ |
| TimeCAP (Lee et al., 2025) | Generic | 📈 | ✗ | ✓ | ✗ | ✓ |
| WeatherBench2 (Rasp et al., 2023) | Weather | 📈 | ✓ | ✗ | ✗ | ✗ |
| Weather2K (Huber et al., 2023) | Weather | 📈 | ✓ | ✗ | ✗ | ✗ |
| ForecastBench (Karger et al., 2024) | Generic | 📄 | ✗ | ✗ | ✓ | ✓ |
| FinBench (Xie et al., 2024) | Finance | 📄 | ✓ | ✓ | ✓ | ✓ |
| Time-MQA (Kong et al., 2025) | Generic | 📄 | ✓ | ✓ | ✓ | ✗ |
| FNSPID (Dong et al., 2024) | Finance | 📈+📄 | ✓ | ✗ | ✗ | ✓ |
| TimeseriesExam (Cai et al., 2024) | Generic | 📈+📄 | ✓ | ✓ | ✓ | ✗ |
| **MTBench (Ours)** | Generic | 📈+📄 | ✓ | ✓ | ✓ | ✓ |

## 3 DATASET COLLECTION & PREPROCESSING

We focus on financial and weather domains for dataset construction due to their high practical relevance and suitability for evaluating LLMs' multimodal integration and reasoning capabilities. In financial markets, linking stock price movements with news sentiment is fundamental for applications such as risk assessment, algorithmic trading, and economic forecasting. In weather, integrating meteorological variables with textual reports is essential for climate monitoring, supply chain planning, and disaster preparedness. Both domains exhibit inherent complexity from dynamic external factors, uncertainty, and event-driven volatility. Importantly, the design principles of our benchmark can

be readily adapted to other application areas, such as healthcare, energy, or transportation, where time-series signals are naturally in conjunction with textual narratives.

### 3.1 DATA COLLECTION AND ALIGNMENT

#### 3.1.1 FINANCE DATASET

The pipeline of finance dataset collection is shown in Figure 2. We collected over 200,000 financial news article URLs from professional financial websites, including *GlobeNews, MarketWatch, SeekingAlpha, Zacks, Invezz, Quartz (QZ), PennyStocks, and Benzinga*, covering the period from May 2021 to September 2023 and offering substantial source diversity. For each URL, we parsed the full content, title, associated stock names, and publication date. From this corpus, we curated a subset of 20,000 articles, ensuring a balanced distribution of article lengths to maintain representativeness. To enrich the dataset with structured metadata, we leveraged GPT-4o (Hurst et al., 2024) to annotate each article with attributes such as content type, temporal effect range, and sentiment. Detailed descriptions of the annotation schema and fine-grained label definitions are provided in Appendix B.1.

**Stock Time-Series Collection**. For each financial news article, we identified the corresponding stock time-series data by utilizing the extracted sentiment and stock name. The historical data was retrieved with stock prices sampled at varying granularities. To ensure data quality, we discarded samples where stock price data was missing for more than 70% of the time period due to market closures (*e.g.,* holidays, weekends). To construct aligned input–output pairs, we anchored each news article at the 0.9 percentile of its input time-series window to capture prior context while preserving predictive relevance. We designed two forecasting horizons: in the **short-term setting**, the model observes the past 7 days of stock prices at a 5-minute resolution to predict movements over the following 1 day; in the **long-term setting**, the model uses 30 days of historical prices at a 1-hour resolution to predict stock movements over the subsequent 7 days.

**Financial Article and Stock Pair**. To align financial news with stock price movements, we matched each article's publication timestamp to the corresponding stock time-series, then compared the article's semantic sentiment with the ground truth stock trend, which is defined as the average price change between input and output windows. Recognizing that not all articles provide reliable signals for future movement, we partitioned the data into two subsets: Consistent Pairs, in which sentiment and trend directions align for approximately 80% of the news–stock pairs, and Misaligned Pairs, where alignment occurs in only 20% of cases. The consistent subset enables evaluation of how effectively LLMs exploit informative textual cues for accurate forecasting, whereas the misaligned subset probes the model's robustness in filtering out misleading or irrelevant news.

#### 3.1.2 WEATHER DATASET

We selected 50 U.S. airports as data sources using the GHCN-H dataset (Menne et al., 2023), covering hourly records from 2003 to 2020. Airports were chosen for their reliable weather measurements, which include temperature, humidity, wind, visibility, pressure, and precipitation. We also incorporated the Storm Events Database (DOC/NOAA/NESDIS/NCDC & National Climatic Data Center, NESDIS, NOAA, U.S. Department of Commerce, 2023), which documents severe U.S. weather events from 1950 to 2020, including storm type, location, and casualties. Each event is tagged with a related phenomenon (*e.g.,* a hurricane spawning multiple tornadoes) and includes narrative descriptions that provide valuable context for reasoning tasks.

**Weather Event Report and Record Alignment**. We aligned storm events with the nearest airport's weather station data within a 50 km radius, grouping events by storm type and merging them into unified records. For some missing narratives in the manually entered storm dataset, we used LLMs to generate synthetic news articles that integrate numeric and textual summaries (see Appendix B.2).

Using each storm's end time as an anchor, we retrieved the preceding 7 days of weather data (inclusive) to forecast the following day's weather. To support richer evaluation, we designed two forecasting horizons: short-term (7→1 day) and long-term (14→3 days). Therefore, each station's 7-day and 14-day time series were paired with the longest news articles to construct the multimodal queries.

## 3.2 Dataset Statistics

### 3.2.1 Dataset Size and Text Length

The finance dataset consists of 20,000 labeled financial news articles paired with time-series data, supporting analysis of market trends, sentiment, and narrative influence. The weather dataset contains 2,000 time-series and synthetic news pairs from 50 U.S. stations (40 pairs each), combining meteorological time-series spanning 7 or 14 days with synthetic news-style narratives. Token length distributions for both the financial corpus and weather samples are provided in Appendix B.

### 3.2.2 Text Duration and Category Distribution

To better characterize the collected datasets, we analyze the distribution of content categories in both financial and weather domains.

**Finance Dataset**. We evaluate the characteristics of the financial news along three dimensions: (1) **Content Type**, categorizing articles as Market News & Analysis, Investment & Stock Analysis, or Trading & Speculative Investments; (2) **Temporal Effect Range**, estimating the expected duration of news impact as Backward-Looking, Present-Focused, or Forward-Looking; and (3) **Sentiment**, assigning polarity based on potential market impact.

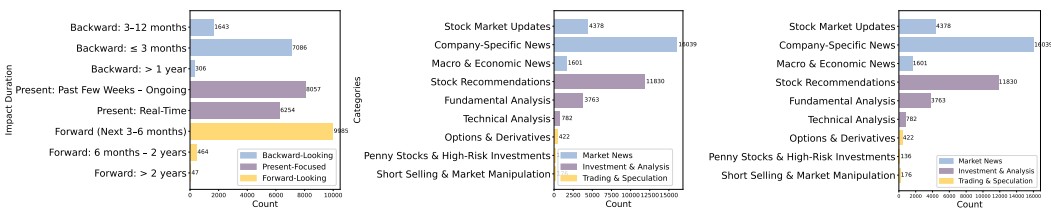

Figure 3: Distribution of content types (left), temporal impact range(middle), and sentiment (right).

The distributions of these labels are shown in Figure 3. The majority emphasize short-term retrospectives or near-term forecasts, which motivates our definition of both short- and long-horizon forecasting tasks. Most articles pertain to company-specific news, aligning naturally with individual stock time series. In addition, most articles convey a positive outlook on stock performance, consistent with the general upward trajectory of the U.S. market between 2021 and 2023.

**Weather Dataset**. Figure 4 presents the distribution of severe weather event types and their durations. The dataset is dominated by short-lived, high-frequency events such as Thunderstorm Winds, Flash Floods, and Hail, with the majority lasting fewer than six hours. This reflects the transient nature of localized atmospheric disturbances. In contrast, long-duration events such as extended storms or heatwaves are comparatively rare. These distributions demonstrate that the dataset captures both immediate fluctuations and extended climate patterns. The inclusion of a broad spectrum of severe events makes the dataset particularly valuable for studying short-term atmospheric variability.

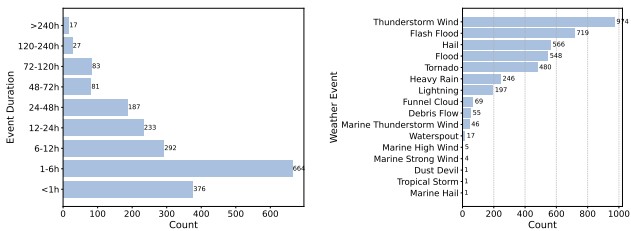

Figure 4: Severe event duration (left) and distribution (right).

## 4 Task Curation

This section presents an overview of the multi-task setup designed for evaluating models across different domains. Each task is structured to assess specific capabilities in forecasting, trend prediction, and complex question answering. Metaprompts used to query LLMs are in Appendix C.

## 4.1 Time-Series Forecasting Task

**Task Setting**. The objective of this task is to forecast future time-series values based on historical observations. To assess the ability of LLMs to integrate multi-modal information, we compare two settings: one using only time-series data and the other combining time-series data with textual news inputs. We evaluate both short-term and long-term forecasting settings. In finance, long-term forecasting is based on 30 days of historical input, reflecting the market's tendency to incorporate longer memory patterns. Conversely, the long-term forecasting in the weather domain adjusts to forecast the next 3 days using the past 14 days of data, accounting for the short-term memory nature of temperature dynamics.

**Evaluation Metrics**. The forecasting task is formulated as a regression problem, with evaluation metrics tailored to the characteristics of each domain. For financial time series, we report Mean Absolute Error (MAE) and Mean Absolute Percentage Error (MAPE) to assess both absolute and relative prediction accuracy. For weather time series, we use Mean Squared Error (MSE) and MAE, which better capture overall deviations and magnitude discrepancies.

## 4.2 Semantic Trend Analysis

To characterize time-series trends, we compute the percentage change between the past and future time series and categorize the results into discrete trend labels. This approach allows us to analyze the directional movement of time-series data and assess the model's ability to classify trends accurately.

**Trend Calculation and Discretization.** For financial time series, the percentage change is defined as the difference between the last and first data points of the output time series, nor-

Table 2: Trend Bin for Financial and Weather Data

| 3-way | 5-way | Finance | Weather |
|---|---|---|---|
| Negative | Bearish Warning | $< -4\%$ $-4\% \sim -2\%$ | Past: $< -0.25$ Future: $< -0.5$ |
| Neutral | Neutral | $-2\% \sim 2\%$ | Past: $-0.25 \sim 0.25$ Future: $-0.5 \sim 0.5$ |
| Positive | Growth Bullish | $2\% \sim 4\%$ $> 4\%$ | Past: $> 0.25$ Future: $> 0.5$ |

malized by the first data point. In contrast, for the weather domain, in past trend analysis, the trend is determined by computing the slope of the daily average temperature over the input days. In future trend prediction, we define the trend as the difference in the daily average temperature of the last day and the future day. To facilitate trend classification, we discretize the computed percentage changes into predefined bins as shown in Table 2. For finance data, we consider both 3-way and 5-way classification, and for weather data, we only consider 3-way classification based on the slope.

## 4.3 Technical Indicator Prediction

To assess the capability of LLMs in predicting domain-relevant metrics, we introduce a technical indicator prediction task, where LLMs forecast key indicators derived from the output time series. These indicators provide higher-level insights into future trends beyond simple price or temperature predictions. In the financial domain, we focus on two widely adopted technical indicators. Moving Average Convergence Divergence (**MACD**) and Upper Band of the Bollinger Bands (**BB**). For the weather domain, we consider indicators closely tied to practical decision-making: next-day maximum and minimum temperature, as well as next-day temperature difference (detailed in Appendix A). All tasks are framed as regression problems and evaluated using MSE and MAE. By emphasizing derived indicators rather than raw values, this task better reflects real-world applications.

## 4.4 News-driven Question Answering

Existing multimodal time-series datasets rarely focus on reasoning-heavy tasks like question answering (QA), limiting their ability to evaluate joint interpretation of text and time-series data. To fill this gap, we introduce a news-driven QA task with two subtasks: correlation prediction and multi-choice QA. As shown in Figure 5, LLMs receive both textual and time-series inputs and must infer their relationship to future trends (see Appendix D.2 for an example in the weather domain).

**Correlation Prediction**. Linking news sentiment to subsequent stock movements is inherently challenging due to market unpredictability. An effective LLM should be able to infer not only

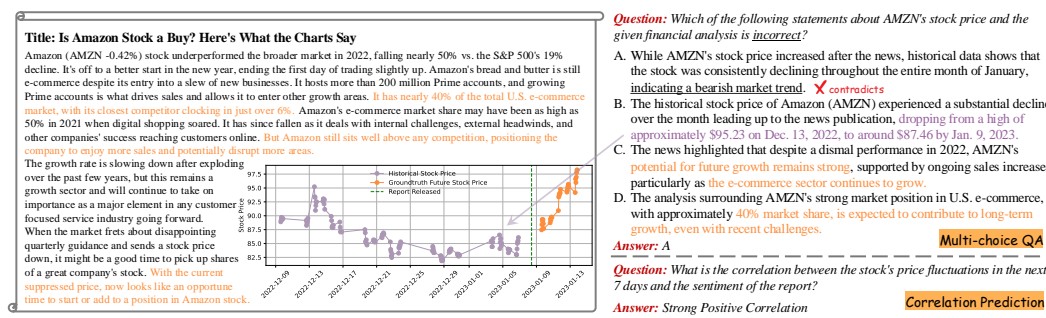

Figure 5: An Example of Multi-choice QA and Correlation Prediction on Finance Dataset

whether such a causal relationship exists but also its direction and strength. We formulate this as a classification task under two labeling schemes: a coarse 3-way classification (*positive*, *neutral*, *negative*) and a finer 5-way classification (*strong/moderate positive*, *no relation*, *moderate/strong negative*). Labels are created with actual price data to ensure real-world alignment. The label distribution (shown in Figure 9 in Appendix B.4) reveals a predominance of negative correlations, which is a slightly surprising finding, as it suggests that in most cases, news sentiment is inversely related to subsequent stock trends. This asymmetry not only complicates the learning signal but also presents a particularly difficult challenge for model performance.

**Multi-choice Question Answering**. This task is designed to evaluate an LLM's ability to reason over multimodal textual analysis with time-series comprehension. To construct this task, we prompt an LLM to generate both correct and incorrect statements based on stock price time-series, as well as accompanying news articles. The correctness of a statement is determined by grounding it in textual evidence from the news, factual trends in the future time series, or valid causal relationships. In contrast, incorrect statements may stem from false claims, misinterpretations of events, flawed causal inferences, or misunderstandings of time-series trends. Once generated, these correct and incorrect statements are formulated into multi-choice QA samples for LLM evaluation. As illustrated in Figure 5, the incorrect statement (A) contradicts the information provided in the financial news, while the correct statements (B, C, and D) can be inferred from either the news content or the stock price movements. This task challenges models to not only comprehend the semantic meaning of textual and numerical time series but also discern causal relationships between them.

# 5 BENCHMARKING AND EVALUATION

## 5.1 EXPERIMENTAL SETTING

**Baseline Models**. MTBench serves as a testbed for evaluating the zero-shot time-series reasoning abilities of LLMs. We benchmark the following models: **GPT-4o** (Hurst et al., 2024), **Claude-Sonnet-3.5-20241022** (The), **Gemini-2.0-Flash** (Team et al., 2023) and **DeepSeek-Chat** (Liu et al., 2024a), with **OpenAI-o3** (OpenAI, 2025) and **LLaMA 3.1-8B** (Touvron et al., 2023) added for select tasks. All models were tested on both time-series-only (denoted as *TS-only*) and time-series+text (denoted as *w/ Text*) settings, except for news-driven QA, which requires context as necessary textual input. This setup enables assessment of how well LLMs integrate structured and unstructured modalities. Traditional time-series models, which lack cross-modal capability, are excluded from the main comparison (see Appendix E.2). Appendix C provides detailed prompts in use for each experiment.

## 5.2 EXPERIMENTAL RESULTS

### 5.2.1 TIME-SERIES FORECASTING

Table 3 shows time series forecasting results for stock and temperature data under TS-only and TS+Text settings. Models perform better in short-term forecasting (*e.g.*, 7-day input, 1-day output), reflecting the difficulty of capturing long-range temporal dependencies. Incorporating text generally improves accuracy, with average gains of 3.6% for stock and 9.8% for temperature forecasting. An illustrative case is provided in Appendix D.1, where text helps correct a failed TS-only prediction. Additionally, we observe that LLMs often fail to generate outputs in the expected format, *e.g.*, 24

hourly temperatures, especially in long-term tasks. We post-process such outputs for fair comparison (detailed in Appendix E.1), but these inconsistencies indicate the need for improved adherence of LLMs to structured output formats in time-series tasks.

Table 3: Forecasting performance under TS-only and TS+Text settings. Left: Stock prices (7-day/30-day). Right: Temperature (7-day/14-day).

| | 7-Day | | | | 30-Day | | | |
|---|---|---|---|---|---|---|---|---|
| | MSE | | MAE | | MSE | | MAE | |
| | TS | w/ Text | TS | w/ Text | TS | w/ Text | TS | w/ Text |
| **GPT-4o** | 1.687 | 1.596 | 0.685 | 2.544 | 2.387 | 2.338 | 3.739 | 3.520 |
| **Gemini** | 1.675 | 1.628 | 3.434 | 3.513 | 2.587 | 2.432 | 3.568 | 3.268 |
| **Claude** | 1.358 | 1.422 | 1.923 | 2.098 | 2.126 | 2.065 | 3.020 | 2.847 |
| **DeepSeek** | 1.753 | 1.720 | 2.085 | 2.135 | 2.357 | 2.134 | 3.482 | 3.305 |
| **OpenAI-o3** | **1.032** | **0.929** | **1.435** | **1.324** | **1.857** | **1.704** | **2.437** | **2.231** |

| | 7-Day | | | | 14-Day | | | |
|---|---|---|---|---|---|---|---|---|
| | MSE | | MAE | | MSE | | MAE | |
| | TS | w/ Text | TS | w/ Text | TS | w/ Text | TS | w/ Text |
| **GPT-4o** | 21.67 | 17.55 | 3.45 | 3.11 | 45.59 | 40.43 | 4.65 | 4.49 |
| **Gemini** | 25.75 | 24.31 | 3.82 | 3.67 | 56.10 | 29.47 | 4.53 | 4.03 |
| **Claude** | 30.34 | 22.48 | 4.11 | 3.50 | **32.01** | 25.08 | **4.24** | 3.75 |
| **DeepSeek** | 31.02 | 29.38 | 4.15 | 4.04 | 61.80 | 101.28 | 5.36 | 6.61 |
| **OpenAI-o3** | **20.68** | **16.14** | **3.35** | **3.09** | 40.57 | **24.97** | 4.42 | **3.51** |

Table 4: Left: Accuracies of Stock trend classification with 3-way and 5-way trend labels on the news-stock pair dataset. Right: Accuracies of past temperature trend classification and future temperature prediction.

| | 7-Day (Acc. %) | | | | 30-Day (Acc. %) | | | |
|---|---|---|---|---|---|---|---|---|
| | 3-way | | 5-way | | 3-way | | 5-way | |
| | TS | w/ Text | TS | w/ Text | TS | w/ Text | TS | w/ Text |
| **GPT-4o** | 40.93 | 42.81 | 34.18 | 36.45 | 34.90 | 47.35 | 19.85 | 30.58 |
| **Gemini** | 41.30 | 47.30 | 34.00 | 41.50 | 37.05 | 44.90 | 21.15 | 29.70 |
| **Claude** | 41.20 | 44.90 | 34.40 | 33.40 | 36.20 | 52.05 | 21.10 | 31.70 |
| **DeepSeek** | 40.53 | 45.12 | 32.85 | 35.60 | 35.50 | 48.26 | 20.70 | 29.55 |
| **OpenAI-o3** | **53.81** | **60.99** | **41.69** | **47.00** | **38.54** | **59.53** | **25.49** | **41.70** |

| | Past (Acc. %) | | Future (Acc. %) | |
|---|---|---|---|---|
| | TS | w/ Text | TS | w/ Text |
| **GPT-4o** | 69.47 | 66.36 | 23.07 | 43.54 |
| **Gemini** | 53.19 | 56.96 | 17.91 | 51.76 |
| **Claude** | 70.44 | 59.78 | 33.23 | 56.87 |
| **DeepSeek** | 22.61 | 26.49 | 16.89 | 25.17 |
| **OpenAI-o3** | **71.58** | **68.42** | **40.52** | **60.79** |

### 5.2.2 SEMANTIC TREND PREDICTION

For stock trend prediction, we evaluate LLMs on their ability to forecast both short-term (*i.e.,* , 7-day input) and long-term (*i.e.,* , 30-day input) stock price movements under 3-way and 5-way classification schemes. To improve reliability, we adopt Chain-of-Thought (CoT) prompting (Wei et al., 2022), which encourages models to articulate intermediate reasoning steps before producing final predictions. For temperature trend classification, models are assessed on their capacity to analyze past dynamics and infer future trends from a 7-day historical temperature series. The results, summarized in Table 4, reveal three key insights. First, models achieve substantially higher accuracy in classifying past trends than in predicting future ones, underscoring the inherent difficulty of forecasting directional changes. Second, incorporating textual data generally improves performance relative to using time-series alone, with consistent gains in 25 out of 28 evaluated cases. Third, an exception emerges in past trend classification, where textual information occasionally reduces accuracy, suggesting that models may not always integrate multimodal context effectively for retrospective analysis.

### 5.2.3 TECHNICAL INDICATOR CALCULATION

Table 5 (a) reports LLM performance on financial indicator prediction, specifically Moving Average Convergence Divergence (MACD) and the upper Bollinger Band (BB), using 7-day and 30-day stock inputs. Incorporating textual input consistently reduces error across models, underscoring the value of contextual information for temporal reasoning. Among all models, OpenAI-o3 achieves the strongest overall performance, particularly in BB prediction. Interestingly, MACD benefits less from textual data than BB, likely because it is more tightly coupled to intrinsic historical price dynamics, whereas BB is more sensitive to external events and volatility captured in textual narratives. Table 5 (b) presents

Table 5: (a) MSE performance of stock technical indicators predictions (upper). (b) MSE performance of Min/Max/Diff temperature prediction (lower)

| | 7-Day | | | | 30-Day | | | |
|---|---|---|---|---|---|---|---|---|
| | MACD | | BB | | MACD | | BB | |
| | TS | w/ Text | TS | w/ Text | TS | w/ Text | TS | w/ Text |
| GPT-4o | 0.430 | 0.365 | 1.450 | 1.082 | 1.003 | 0.897 | 2.521 | 2.068 |
| Gemini | 0.482 | 0.384 | 1.280 | 1.153 | 1.132 | 0.975 | 2.565 | 2.248 |
| Claude | **0.241** | 0.373 | 2.105 | 1.246 | 0.970 | 1.171 | 2.605 | 2.345 |
| DeepSeek | 0.435 | 0.352 | 1.526 | 1.187 | 1.053 | 1.072 | 2.486 | 2.201 |
| OpenAI-o3 | 0.384 | **0.246** | **1.025** | **0.687** | **0.823** | **0.586** | **2.015** | **1.523** |

| | Maximum | | | | Minimum | | | | Difference | | | |
|---|---|---|---|---|---|---|---|---|---|---|---|---|
| | MSE | | MAE | | MSE | | MAE | | MSE | | MAE | |
| | TS | w/ Text | TS | w/ Text | TS | w/ Text | TS | w/ Text | TS | w/ Text | TS | w/ Text |
| Llama3.1 | 37.56 | 33.87 | 4.67 | 4.42 | 21.21 | 18.80 | 3.44 | 3.22 | 65.77 | 54.28 | 6.54 | 5.85 |
| GPT-4o | 26.03 | 19.58 | 3.76 | 3.02 | 15.58 | 15.39 | 2.89 | 2.76 | 27.06 | **18.84** | 3.86 | **3.20** |
| Gemini | 25.98 | **16.39** | 3.77 | **2.96** | 16.20 | 16.27 | 2.94 | 2.93 | 35.72 | 23.21 | 4.40 | 3.63 |
| Claude | **23.18** | 18.69 | **3.59** | 3.21 | **14.57** | **13.42** | **2.73** | **2.63** | **21.03** | 19.10 | **3.41** | 3.26 |
| DeepSeek | 33.90 | 32.82 | 4.45 | 4.38 | 18.39 | 17.25 | 3.16 | 3.05 | 49.28 | 44.99 | 5.51 | 5.24 |

than BB, likely because it is more tightly coupled to intrinsic historical price dynamics, whereas BB is more sensitive to external events and volatility captured in textual narratives. Table 5 (b) presents

an evaluation of LLMs in predicting next-day maximum, minimum, and temperature differences based on a 7-day historical temperature. Across all metrics, incorporating textual data alongside time-series generally improves performance, as reflected in reduced MSE and MAE values. Notably, temperature difference prediction is most challenging, exhibiting higher errors relative to other prediction tasks. The overall trend highlights the importance of multimodal learning in time-series forecasting, where textual context can enhance predictive accuracy, but effectiveness still depends on the model's inherent capability to process numerical and textual data synergistically.

### 5.2.4 NEWS-DRIVEN QUESTION ANSWERING

Table 6 shows the results of LLMs on predicting the correlation between news and future stock fluctuations, and their performance on multiple-choice QA (MCQA) in finance and weather domains. We make several observations from the results. Firstly, long-term stock correlation prediction is less challenging, as LLMs consistently perform better in the 30-day setting compared to the 7-day setting. This suggests that short-term stock movements are more unpredictable due to random market noise and external events, whereas long-term correlations between financial news and stock prices are more stable and easier for models to capture. Moreover, short-term MCQA is generally easier than long-term MCQA. This indicates that models can effectively leverage recent information when answering questions but struggle with reasoning over longer time horizons, where knowledge decay or compounding factors might introduce additional complexity. Overall, the results emphasize the strengths of LLMs in reasoning tasks while exposing their limitations in handling the randomness of short-term financial fluctuations.

Table 6: Accuracy Comparison on Different Tasks

| | News-stock Correlation | | | | News-driven MCQA | | | |
| | 7-Day | | 30-Day | | 7-Day | | 30-Day | |
| | 3-way | 5-way | 3-way | 5-way | Finance | Weather | Finance | Weather |
|---|---|---|---|---|---|---|---|---|
| Gemini | 51.8 | 26.4 | **59.6** | 34.8 | 63.6 | 43.4 | 50.3 | 54.0 |
| Claude | 50.4 | 29.0 | 57.9 | 34.3 | 75.6 | **51.8** | 61.1 | 51.2 |
| GPT-4o | **53.6** | **31.0** | 57.6 | 34.6 | 65.1 | 41.7 | 52.8 | 44.8 |
| DeepSeek | 50.0 | 27.1 | 57.5 | **35.0** | 77.6 | 46.7 | **69.3** | **57.3** |

Figure 6: Correlation Confusion Map

Figure 6 shows the confusion map of different LLMs predicting the correlation between news and stock price movements in a 5-way classification setting. A key observation is that models (*e.g.,* GPT-4o, Gemini) exhibit a strong bias toward classifying news-stock pairs as having a moderate positive correlation, regardless of external factors or market conditions that might affect stock price movement. This suggests an inherent assumption or systematic limitation in LLMs, where they struggle to capture the full spectrum of correlation dynamics and instead default to a middle-ground prediction. The models thereby fail to properly analyze negative or weak correlations, possibly due to difficulties in disentangling causal relationships between textual and numerical data, which highlights a broader challenge in using LLMs for financial forecasting.

## 6 CONCLUSION AND FUTURE WORK

We present MTBench, a large-scale benchmark for evaluating LLMs' ability to reason over multi-modal time-series and text in finance and weather domains. MTBench emphasizes semantic and temporal alignment between numerical trends and narrative information, supporting diverse tasks such as forecasting, trend analysis, and QA. Our results show that while LLMs demonstrate promise, they continue to struggle with long-range temporal reasoning, causal inference, and multimodal integration. Textual input often improves performance, though gains are uneven, and models frequently fail to produce well-structured outputs in long-term settings.

MTBench focuses on financial and weather data but can be naturally extended to domains such as healthcare and social sciences, where temporal reasoning is equally critical. Beyond zero-shot evaluation, future directions include fine-tuning, hybrid architectures, and temporally aware training objectives, as well as handling multivariate and multi-resolution time-series, incorporating additional modalities (*e.g.,* images), and detecting misaligned or misleading context. Addressing persistent limitations—such as output inconsistency and correlation bias—remains essential for developing robust, generalizable systems.

**Reproducibility Statement**. To ensure reproducibility, we provide the complete evaluation scripts and all associated code in the supplementary material. Upon acceptance, we will publicly release the benchmark to support transparent and fair evaluation within the research community. For all LLMs evaluated in our study, we explicitly specify the model versions and access configurations, allowing experiments to be replicated under the same conditions. Together, these efforts are intended to facilitate rigorous verification of our results and to establish a standardized foundation for future research on multimodal time-series reasoning.

**LLM Usage Statement.** In preparing this submission, we used large language models (LLMs) exclusively for polishing the writing style, including improving grammar, fluency, and readability. All technical content, experimental design, implementation, analysis, and conclusions were developed entirely by the authors without using LLMs.

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

# Appendix

## A  DETAILS OF TECHNICAL INDICATOR PREDICTIONS

For finance data, we adopt two widely used technical indicators. **Moving Average Convergence Divergence (MACD)**. MACD is calculated as the difference between the 12-day and 26-day exponential moving averages (EMAs) of the stock price. It helps identify momentum shifts and trend reversals. The model is tasked with predicting the MACD values for the forecasted time period.

**Upper Band of the Bollinger Bands**. Bollinger Bands are volatility-based indicators consisting of an upper band, a lower band, and a moving average. The upper band is defined as Upper Band = SMA + $k \cdot \sigma$, where SMA is the simple moving average over a defined window, $\sigma$ is the standard deviation of prices over the same window, and $k$ is a constant (typically 2). This indicator helps assess volatility and potential overbought conditions.

For weather data, we adopt the following indicators.

**Next-Day Maximum & Minimum Temperature**. Given past temperature data, the model predicts the highest and the lowest temperature for the next day.

**Next-Day Temperature Difference**. Given past temperature data, the model predicts the difference between the maximum and minimum temperatures for the next day.

## B  DATASET DETAILS

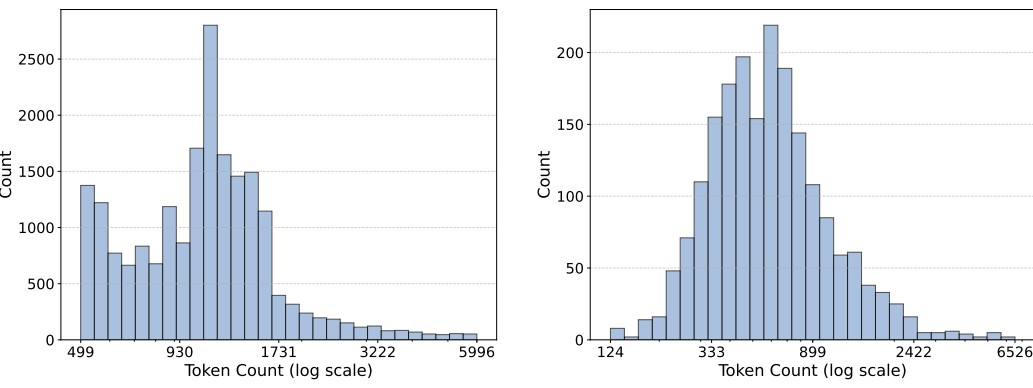

Figure 7: Article token counts distribution of financial (left) and weather (right) dataset.

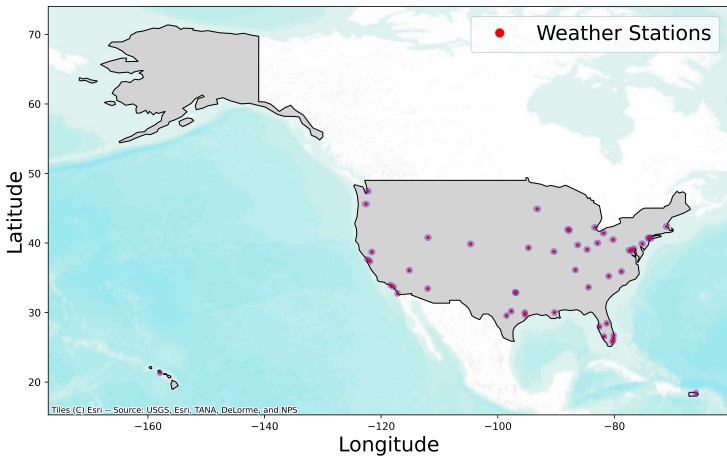

Figure 8: Geographical distribution of weather stations

## B.1 FINE-GRAINED FINANCIAL NEWS LABEL

**Categorization 1: News Type**. Select the most relevant category and subcategory.

1. Market News & Analysis

   (a) Macro & Economic News: Covers broader economic trends, interest rates, inflation, and geopolitical events.
   (b) Stock Market Updates: Daily/weekly overviews of market indices, sector performance, and notable stock movements.
   (c) Company-Specific News: Earnings reports, mergers & acquisitions, leadership changes, or major corporate announcements.

2. Investment & Stock Analysis

   (a) Fundamental Analysis: Examines a company's financial health using earnings, revenue, P/E ratio, etc.
   (b) Technical Analysis: Uses chart patterns and indicators (e.g., moving averages, RSI) to predict stock movements.
   (c) Stock Recommendations: Buy/sell/hold ratings, analyst upgrades/downgrades, and price target projections.

3. Trading & Speculative Investments

   (a) Options & Derivatives: Strategies for options trading, futures, and leveraged instruments.
   (b) Penny Stocks & High-Risk Investments: Coverage of micro-cap stocks and speculative assets.
   (c) Short Selling & Market Manipulation: Insights on short squeezes, pump-and-dump schemes, and regulatory issues.

**Categorization 2: Temporal Impact**. Select all applicable labels.

1. Backward-Looking (Retrospective Analysis)

   (a) Short-Term Retrospective ($\leq$ 3 months): Covers recent earnings reports, economic data releases, and short-term market performance.
   (b) Medium-Term Retrospective (3–12 months): Analyzes company performance over the last fiscal year, sector trends, and regulatory changes.
   (c) Long-Term Retrospective ($>$ 1 year): Historical financial analysis, decade-long economic cycles, and structural market shifts.

2. Present-Focused (Current Market Insights)

   (a) Real-Time Market Developments: Covers breaking news, stock price movements, and intraday financial events.
   (b) Recent Trends (Past Few Weeks – Ongoing): Tracks market sentiment, investor behavior, and economic conditions shaping current investment decisions.

3. Forward-Looking (Forecasting & Projections)

   (a) Short-Term Outlook (Next 3–6 months): Includes earnings guidance, analyst predictions, and upcoming economic events.
   (b) Medium-Term Outlook (6 months – 2 years): Covers strategic corporate decisions, macroeconomic forecasts, and sectoral growth trends.
   (c) Long-Term Outlook ($>$ 2 years): Encompasses structural investment themes, demographic shifts, and innovation-driven disruptions.

**Categorization 3: Sentiment**. Select the most appropriate label.

1. Positive Sentiment

   (a) Bullish: Optimistic outlook on a stock, sector, or market.
   (b) Growth-Oriented: Highlights expansion, revenue increase, or new opportunities.

      (c) Upbeat Market Reaction: Positive investor sentiment driven by earnings beats, regulatory approvals, or strong guidance.

2. Neutral Sentiment

      (a) Balanced/Informational: Presents data or events objectively.

      (b) Mixed Outlook: Covers both positive and negative factors, leading to uncertainty.

      (c) Speculative: Discusses potential future scenarios without strong directional bias.

3. Negative Sentiment

      (a) Bearish: A pessimistic outlook, indicating expected declines or underperformance.

      (b) Risk & Warning: Highlights financial risks, regulatory threats, or economic downturns.

      (c) Market Panic/Fear: Reports significant uncertainty, volatility, or investor anxiety.

## B.2 SYNTHETIC WEATHER REPORT

For cases where narrative descriptions are missing in the original storm dataset, we use LLMs to generate synthetic news articles based on the merged event record. An example is shown as follows:

> The following events were reported: Tornado. These occurred near station USW00012842, approximately 38.118 km away, between 2019-10-18 20:29:00 and 2019-10-18 22:28:00.Thankfully, no injuries or fatalities were reported. The events caused property damage valued at 10100000.0 and crop damage of 0.0. Episode Narrative: Tropical Storm Nestor developed in the Gulf of Mexico and moved northeast, making landfall on St. Vincent Island in the Florida Panhandle on the afternoon of the 19th. The bulk of the convection developed on the eastern and southeastern side of the storm, with a couple bands of showers and storms moving into the west coast of the Florida Peninsula. These bands of storms produced 3 confirmed tornadoes, including one EF-2 tornado in Polk County. Otherwise, straight line winds were minimal in west central and southwest Florida and caused little impacts. The highest storm total rainfall in west central and southwest Florida was 7.77 inches in Baskin in Pinellas County, with other areas in Pinellas and parts of Hillsborough County seeing 5 to 6 inches, causing minor nuisance flooding. The highest storm surge in the area was 3.6 feet at Cedar Key. Taking into account the astronomical tide cycle, this resulted in a peak water level of 2.27 feet MHHW at 5:18 AM EDT on the 19th. Tropical Storm Nestor developed in the Gulf of Mexico and moved northeast, making landfall on St. Event Narrative: Damage was reported to several homes in the Twelve Oaks Mobile Home Park in Seminole. A few homes had roof, window, and carport damage, and several trees were knocked down. No injuries were reported. A long, continuous path of damage was found in western Polk County from a tornado, causing extensive EF-2 damage. An NWS survey and subsequent Civil Air Patrol aerial survey found numerous homes and businesses with damage to roofs, fascia, awnings, and screen enclosures, as well as fences and trees knocked down. One home was completely destroyed. Kathleen Middle School sustained significant roof damage, with rain water and sprinkler systems causing additional water damage. A camper was lifted into a residence near the middle school. The tornado crossed Interstate 4, overturning a tractor trailer.

## B.3 GLOBAL DATA STATISTICS

To understand the characteristics of our temperature dataset, we provide summary statistics, including the mean, standard deviation, minimum, and maximum values over the entire dataset in Table 7.

Table 7: Global Statistics of Temperature Data

| Input length | Mean (°C) | Std Dev (°C) | Min (°C) | Max (°C) |
|---|---|---|---|---|
| 7 days | 20.13 | 8.04 | 46.7 | -20.0 |
| 14 days | 19.80 | 8.33 | 46.7 | -20.0 |

Our raw data includes multiple channels besides temperature, such as humidity, wind speed, etc. The detailed statistics of all included features are given in Table 8. Each of these environmental variables provides critical contextual information that complements the primary temperature readings. The

multi-channel nature of our raw dataset enables more robust analysis and modeling by capturing the complex interactions between different atmospheric factors. We leave these for future work.

Table 8: Global Statistics of All Included Features

| Feature | Mean | Std Dev | Min | Max |
|---|---|---|---|---|
| Relative Humidity(%) | 68.02 | 20.79 | 0.00 | 100.00 |
| Station Level Pressure(hPa) | 994.62 | 35.08 | 111.00 | 1352.30 |
| Sea Level Pressure(hPa) | 1016.45 | 6.62 | 960.80 | 1059.90 |
| Wind Speed(m/s) | 3.75 | 2.45 | 0.00 | 439.10 |
| Visibility(km) | 14.27 | 4.01 | 0.00 | 175.42 |
| precipitation (3 hours) (mm) | 1.86 | 5.25 | 0.00 | 763.00 |
| precipitation (6 hours) (mm) | 1.74 | 5.56 | 0.00 | 284.00 |
| precipitation (24 hours)(mm) | 8.82 | 15.06 | 0.00 | 817.80 |

### B.4 CORRELATION DISTRIBUTION

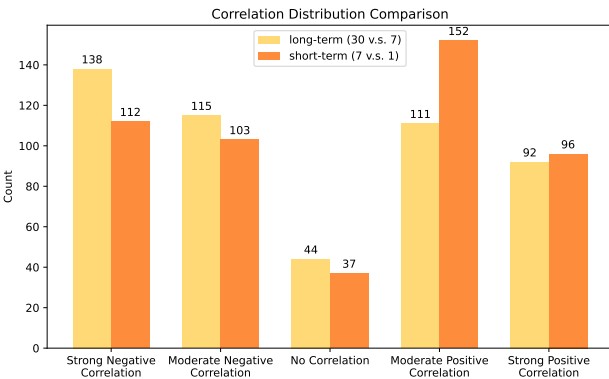

Figure 9: Distribution of correlation between future stock price fluctuation and news sentiment over 500 samples for each input length.

Figure 9 shows the correlation distribution of financial news and stock price fluctuations across both short-term and long-term horizons. Notably, a significant portion of stock-news pairs exhibit a negative correlation, suggesting that financial news does not always align with immediate market movements and, in some cases, may inversely influence investor behavior.

For short-term correlations on a 7-day input, there is a higher frequency of moderate and strong positive correlations, implying that in the immediate aftermath of news publication, market sentiment and reactions often align with the sentiment of the news. However, in the long-term scenario with 30-day input, we observe an increased presence of strong and moderate negative correlations, indicating that initial market reactions may be transient, and other external factors such as macroeconomic trends, delayed investor responses, or broader market corrections play a more dominant role in shaping price movements. This trend highlights a fundamental challenge for LLMs in correlation prediction. It underscores the necessity for models to incorporate a deeper understanding of economic context beyond surface-level sentiment analysis.

## C TASK PROMPT

### C.1 INPUT TYPES

We test on two different input types: Timeseries-only and Timeseries-Text. The prompt examples on the finance dataset are as follows. Time series data is formulated in a list of timestamped values. For the Timeseries-only input type, models receive only the numerical time series data without any

textual context. For the Timeseries-Text input type, models receive both the time series data and accompanying textual descriptions or analyses.

---

**Stock Timeseries-only Trend Prediction**

You are a financial prediction expert with knowledge of advanced machine learning models and time-series analysis. Your goal is to predict the stock trend with given labels based on the following input:
**Time Series Stock Price Data**: This data includes stock prices recorded at 1-hour intervals over the last month from {timestamps[0]} to {timestamps[-1]}.
Example data format: {time_series_data}
**Output**: Provide a prediction for the stock trend and categorize it into one of the following labels:
"<-4%", "-2% -4%", "-2% +2%", "+2% +4%", ">+4%".
**Task**: Please think step-by-step, Analyze the provided time-series data to identify trends and patterns that could impact stock performance. Focus solely on the time-series data for making predictions. Then wrap your final answer in the final predicted label in the format {label}

---

**Stock Timeseries-Text Combined Trend Prediction**

You are a financial prediction expert with knowledge of advanced machine learning models and time-series analysis. Your goal is to predict the stock trend (rise, neutral, or fall) based on the following inputs:
**Time Series Stock Price Data**: This data includes stock prices recorded at 1-hour intervals over the last month from {timestamps[0]} to {timestamps[-1]}. Example data format: {time_series_data}
**News Data**: This includes news headlines and summaries relevant to the stock's company or sector. Example data format: {text}
**Output**: Provide a prediction for the stock trend categorized one of the following labels:
"<-4%", "-2% -4%", "-2% +2%", "+2% +4%", ">+4%".
**Task**: Analyze the provided time-series data and news to identify future trends of the stock performance. Ensure that the news data is used to supplement the insights from the time-series analysis, focusing on combining both inputs for a more accurate prediction.
Please think step-by-step and briefly explain how the combination of time-series data and news data led to the prediction. Then wrap your final answer in the final predicted label in the format {label}

---

## C.2 TASK INSTRUCTIONS

System prompts are useful for establishing model behavior and guiding model outputs to align with specific use cases. We provide the prompts that were used to instruct the models in our experiments on finance dataset and weather dataset as follows.

---

**Stock News-driven QA Prompt**

You are an expert in finance and stock market analysis.
**Correlation Prediction:**
Based on the given 30-day historical stock price time-series and a financial analysis published at the last timestamp of the time-series, your task is to predict the correlation between the stock's price fluctuations in the next 7 days and the analysis sentiment (positive correlation indicates that positive analysis leads to price increase and negative analysis leads to price decrease). Take into account external factors or market conditions that might affect stock price movement.
**Multi-choice QA:**
Your task is to answer the question based on the given 30-day historical stock price time-series and a financial analysis published at the last timestamp of the time-series. Return your answer only in the letter (A, B, C, or D).

---

> **Weather Indicator Prediction Prompt**
>
> You are a weather forecasting AI. The input time-series represents temperature readings. This data is from a location in the United States, where summers are hot and winters are cold. Weather conditions can also be affected by storms, heavy rain, and cold fronts. The daytime is usually warmer than the nighttime. Every 24 temperature readings represent a full day from 00:00 to 23:00. Your task is to analyze the past 7 days' temperature trend and predict the next 1 day's highest temperature and lowest temperature as well as the temperature difference between the highest and lowest temperature.

## D    DETAILED EXAMPLES

### D.1    FINANCIAL TREND PREDICTION

> **Stock Trend Prediction Query**
>
> **Financial Article:**
> **Title: Kandi Technologies Receives Letter of Intent from Coleman Powersports to Purchase 4,800 Crossover Golf Carts in the First Quarter 2023** Dec. 19, 2022 – Kandi Technologies Group, Inc. (the "Company," "we" or "Kandi") (NASDAQ GS: KNDI), today announced that its wholly owned subsidiary SC Autosports has received a letter of intent from Coleman Powersports to purchase 4,800 crossover golf carts in the first quarter of 2023, with a total value of approximately $27.6 million
> **Stock Time Series:**
> This data includes stock prices recorded at 1-hour intervals over the last month from {timestamps[0]} to {timestamps[-1]}. Visualizations are as follows for reference.
>
> 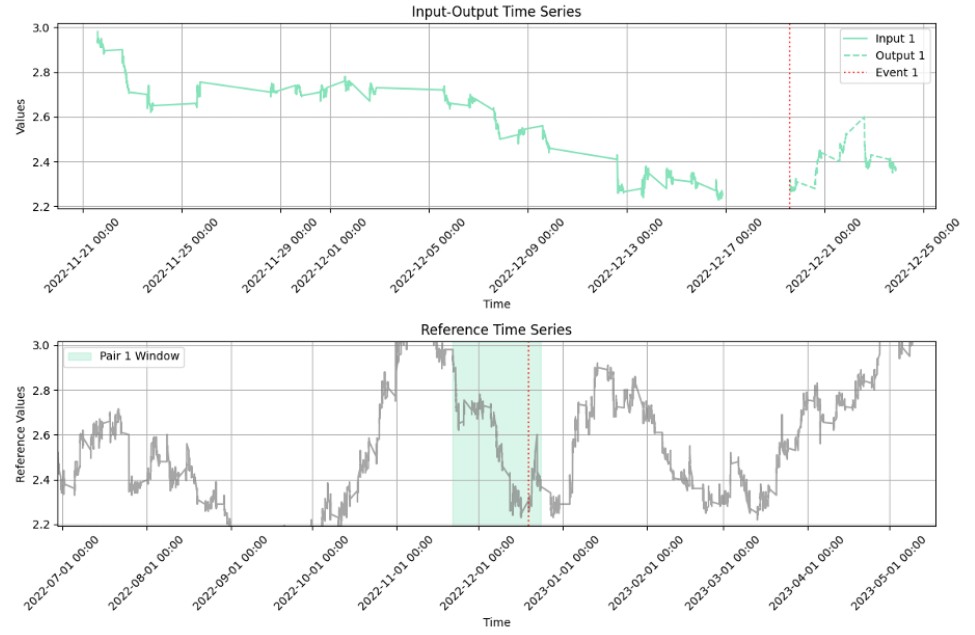
>
> **Question:**
> The trend is defined by one of the following labels: $< -4\%$; $-2\% \sim -4\%$; $-2\% \sim +2\%$; $+2\% \sim +4\%$; $> +4\%$. Analyze the news and the time-series data together to predict the trend.

### GPT-4o Output with Time Series Input

To predict the stock trend using the given time-series data, we need to analyze the data to understand its trend and potential future movement. Here's a step-by-step approach:

1. **Data Overview**: The data starts with higher values and generally decreases over time.

2. **Trend Analysis**: A visual inspection suggests that the overall trend is downward. The prices show a decline from around 2.98 to approximately 2.26.

3. **Percentage Change Calculation**:
   - Initial Price: 2.98
   - Final Price: 2.26

4. **Categorizing the Trend**: Based on the calculated percentage change, the percentage change of approximately -24.5% falls well below the $<$-4% category.

Given this analysis, **the predicted stock trend label is:** $< -4\%$ ✖

### GPT-4o Output with Time Series + Text Input

We'll analyze both the time-series stock price data and the news data, combining insights from both to forecast the future trend.

1. **Time Series Data Analysis:** The stock prices provided show a general declining trend over the last month. The overall decrease from the beginning to the end of the time-series is approximately 24.16% ((2.26 - 2.98) / 2.98 * 100)......

2. **News Data Analysis:** The news release is positive, detailing a significant potential order from Coleman Powersports for 4,800 crossover golf carts valued at $27.6 million. This indicates anticipated growth and increased revenue in the near future, particularly as it suggests a rising trend in orders.....

3. **Combining Insights:** While the time-series data indicates a recent downward trend, the positive news about future orders could counteract this negativity by boosting investor sentiment and potentially leading to a stock price increase. The news is fresh and could have a strong impact as it suggests future revenue growth, which might not yet be reflected in the time-series data. Investors may react positively to the news, expecting future growth, which could lead to a stock price increase in the short term.

**Prediction:** Given the positive news and its potential impact on market sentiment, alongside the current downward trend in the time-series, **final Prediction is** $+2\% \sim +4\%$**.** ✔

### D.2 WEATHER QUESTION ANSWERING EXAMPLES

### Weather QA Examples

**Weather Report:**
A thunderstorm wind and tornado event occurred near station USW00093738 on April 6, 2017, between 11:50 AM and 12:38 PM. No injuries, fatalities, or significant property damage were reported. A low-pressure system over the Ohio Valley created unstable atmospheric conditions, leading to severe thunderstorms and isolated tornadoes. Numerous trees and power lines were downed across multiple locations, with damage reported in Herndon, Warrenton, Airlie, New Baltimore, and Sterling Park. The National Weather Service confirmed multiple EF-0 tornadoes with estimated wind speeds up to 85 mph, causing extensive tree damage, minor structural damage, and power outages. Some trees fell on vehicles and buildings, and a greenhouse and outbuildings were destroyed. The most intense damage occurred along Airlie Road and Beverlys Mill Road. The National Weather Service conducted surveys with assistance from local emergency management agencies.
**Temperature Time Series:**

This data includes temperature recorded at 1-hour intervals over the last 7 days from {timestamps[0]} to {timestamps[-1]}. Visualizations are as follows for reference.

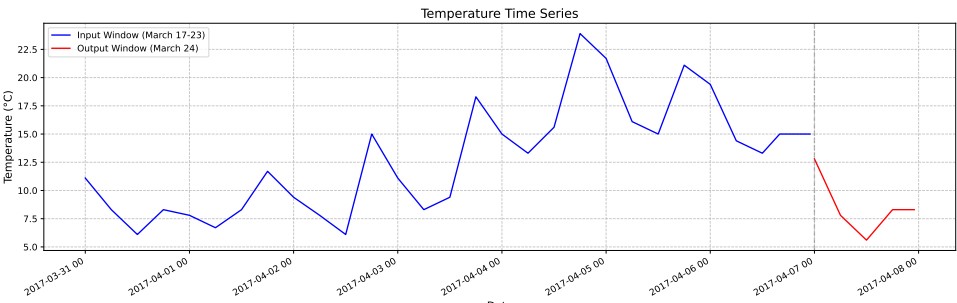

### Indicator Prediction:
**Question:** What will be tomorrow's maximum temperature, minimum temperature and temperature difference? **Answer: Maximum temperature is 12.8. Minimum temperature is 5.6. Temperature difference is 7.2.**

### Trend Prediction:
**Question:** Based on the given information, predict the temperature trend for the next 1 day. Calculate the mean temperature of the last 24-hour period (i.e., the most recent day in the input) and compare it with the mean temperature of the first predicted day. If the difference is greater or equal than 0.5, classify the trend as '*increasing*'. If the difference is less or equal than -0.5, classify the trend as '*decreasing*'. Otherwise, classify it as '*stable*'.
**Answer: decreasing**

### Multi-choice QA:
**Question:** Based on the reported weather events and the temperature trends leading up to the storm on April 6, 2017, which of the following statements is most logically valid regarding the weather phenomenon and its impact on the next day's temperature?
A. The substantial drop in temperatures observed on April 7 is a direct result of the intense thunderstorms and tornadoes that occurred the previous day, creating a cooling effect due to updrafts.
B. The tornadoes were primarily caused by a sudden increase in high pressure which unexpectedly stabilized the region, leading to a warmer day on April 7.
C. The warm and moist air brought by the cutoff low pressure system contributed to the severe weather conditions, but this same system caused a sudden cold front that resulted in lower temperatures by April 7.
D. The warm temperatures during the days preceding the thunderstorm event indicate that the tornadoes were unlikely to produce any significant cooling effects, so April 7 would naturally continue to feature warmer temperatures.
**Answer: A**

## E    ADDITIONAL RESULTS

### E.1    POST-PROCESSING

Since the LLM occasionally produces outputs that deviate from the expected sequence length specified in the prompt (e.g., the target length is 72 timesteps, but the output may contain 69 or 78), we apply the following post-processing methods:

- **Truncation**: If the output exceeds the expected length, we truncate it to the first 72 timesteps.

- **Interpolation**: If the output is shorter than the expected length, we apply linear interpolation to resample the prediction to the desired number of timesteps. This helps maintain temporal smoothness and avoids introducing artificial discontinuities.

## E.2 TIME-SERIES FOUNDATION MODELS FOR FORECASTING

To fairly evaluate the time-series forecasting capabilities of LLMs, which are not fully fine-tuned on our dataset, we compare them against a suite of time-series foundation models (TSFMs) that also do not rely on full-shot training. Specifically, we consider the following TSFM families: **Chronos** (Ansari et al., 2024), **Moirai** (Woo et al., 2024), **TimesFM** (Das et al., 2024), **Time-MoE** (Shi et al., 2024), and **Timer-XL** (Liu et al., 2024c).

When prompting LLMs for forecasting, we intentionally do not normalize the input data, as our goal is to assess whether the model can reason about the values similarly to how humans would, without relying on engineered preprocessing. In contrast, for the time-series foundation models, we normalize the input data before feeding it into the models and then de-normalize the outputs before computing evaluation metrics. This standard practice ensures the numerical stability of training and fair comparison of performance.

The complete forecasting results for these models are presented in Table 9.

Table 9: Forecasting performance of time series foundation models

| | Finance | | | | Weather | | | |
| | 7-Day | | 30-Day | | 7-Day | | 14-Day | |
| | MAE | MAPE | MAE | MAPE | MAE | MAPE | MAE | MAPE |
|---|---|---|---|---|---|---|---|---|
| **Time-MOE-50M** | 8.2118 | 0.2605 | 19.7889 | 1.5731 | 15.1045 | 2.8789 | 17.6383 | 3.1555 |
| **Time-MOE-200M** | 7.6231 | 0.2025 | 17.0126 | 1.5606 | 15.2657 | 2.8940 | 17.6878 | 3.1670 |
| **Chronos-small** | 4.2030 | 0.2433 | 4.9204 | 0.0780 | 21.1943 | 3.3683 | 30.8115 | 3.9502 |
| **Chronos-base** | 3.4189 | 0.0731 | 5.7405 | 0.1997 | 20.7608 | 3.3371 | 28.7851 | 3.8721 |
| **Chronos-large** | 5.2218 | 0.2335 | 5.5264 | 0.1862 | 21.3764 | 3.4131 | 27.7671 | 3.8829 |
| **Moirai-small** | 8.6850 | 0.4880 | 17.7745 | 0.7682 | 73.9020 | 3.8336 | >100 | 21.5944 |
| **Moirai-base** | 9.4369 | 0.5189 | 8.8144 | 0.3108 | >100 | 3.7512 | 4.3122 | >100 |
| **Moirai-large** | 6.5261 | 0.2284 | 16.9172 | 0.4421 | 66.0836 | 3.8193 | 49.9748 | 4.3005 |
| **TimesFM** | 2.4711 | 0.0476 | 5.1919 | 0.0658 | 14.3679 | 2.8191 | 16.0583 | 3.0063 |
| **Timer-XL** | 15.6546 | 0.0691 | 11.3792 | 0.4390 | 14.0215 | 2.7566 | 15.6546 | 2.9547 |

## F DISCUSSION

**Limitations**. While MTBench offers a rich and semantically aligned benchmark across finance and weather domains, it currently focuses on two specific modalities: univariate textual descriptions and structured time-series signals. This scope, while practical, may not fully capture the complexity of real-world multimodal scenarios that involve additional modalities such as tabular financial indicators, satellite imagery, or geospatial data. Additionally, automatic alignment quality may still vary across samples, introducing some noise that could potentially affect fine-grained analysis.

**Future Work**. A natural next step is to expand MTBench to encompass more domains (*e.g.,* healthcare, energy systems), and richer modalities (*e.g.,* images, graphs). Furthermore, the benchmark could be extended to support interactive evaluation protocols (*e.g.,* , user feedback loops, adaptive prompting) and model auditing tools for detecting hallucinations, spurious correlations, or semantic mismatches between modalities. Incorporating more fine-grained annotations and challenge sets, such as adversarial or counterfactual samples, can also enhance its diagnostic value.

**Broader Impact**. By enabling a more rigorous evaluation of LLMs' multimodal reasoning capabilities, MTBench lays the testbed for building trustworthy, context-aware AI systems for time series analysis across critical domains. For instance, better alignment between textual narratives and numerical data can support more accurate financial risk assessments or more timely climate warnings. At the same time, our benchmark can foster research on identifying and mitigating harmful hallucinations or misleading interpretations that may arise from poorly aligned or noisy multimodal inputs, promoting responsible deployment of LLMs in decision-critical environments.

