# OpenReview forum: "MTBench: A Multimodal Time Series Benchmark for Temporal Reasoning and Question Answering"
_ICLR.cc/2026/Conference — ICLR 2026 Conference Desk Rejected Submission_

### Official Review · Reviewer_Qm4M · 2025-10-29

**Soundness:** 3
**Presentation:** 2
**Contribution:** 2
**Rating:** 4
**Confidence:** 4

**Summary:**

MTBench is a benchmark for evaluating multimodal reasoning over time-series data and text. It aligns financial news with stock prices and weather event reports with meteorological data, covering both short and long horizons. The benchmark defines four tasks: forecasting, trend analysis, technical-indicator prediction, and question answering, testing models on numerical prediction and textual reasoning. Experiments with large language models show that text improves short-term accuracy, but long-term temporal reasoning and causal understanding remain difficult.

**Strengths:**

- Real narrative and time-series alignment in two domains (finance and weather), plus Consistent vs. Misaligned splits, dual horizons/resolutions, and MCQA items that require both modalities—useful for testing whether models use helpful text and ignore misleading text.
- The QA tasks (correlation prediction and MCQA) are novel and challenging, requiring reasoning across textual and temporal evidence. They move beyond numeric forecasting to assess causal understanding, consistency checking, and cross-modal inference, which many prior benchmarks lack.

**Weaknesses:**

- For semantic trend analysis and technical-indicator prediction, labels are deterministically derived from the time series; text is optional. Strong results here don’t necessarily demonstrate text-based reasoning, so the benchmark may not isolate the contribution of the text modality.
- Table 1 appears to conflate task type with input modality. It lists Time-MMD as “time-series only” under Query Format, but Time-MMD’s evaluation uses text plus time series inputs (via its multimodal setup). This could mislead readers about the true input requirements.
- The evaluation pipeline feels too trivial for a benchmark—mainly serializing the time series and appending text into prompt templates for general LLMs, with format-fixing post-processing—so scores may reflect prompt-following rather than genuine multimodal temporal reasoning.

**Questions:**

Please address the identified weaknesses and limitations noted above.

---

> ### Author Response · Authors · 2025-11-21
> **Initial Rebuttal to Reviewer Qm4M**
>
> We thank the reviewer for the thoughtful feedback and for highlighting the novelty and clarity of MTBench. Below we address each weakness and limitation directly.
>
> ---
> ## W1:  Labels for semantic trend / technical-indicator prediction come directly from the TS; text is optional.
>
> This behavior is intentional. These tasks are designed not to force reliance on text, but to evaluate **how models behave when text is helpful, unhelpful, or misleading**. Using deterministic TS-derived labels allows controlled counterfactual testing:
>
> - When text is consistent with the TS, a strong model should benefit from it.
> - When text contradicts the TS, a strong model should down-weight or ignore it.
> - When text is irrelevant, robust models should default to TS-only reasoning.
>
> Thus, “text is optional” is a feature of the task design: it enables systematic evaluation of **cross-modal alignment, selective attention, and resistance to narrative bias**—capabilities not captured by pure forecasting benchmarks.
> ## W2: Table 1 conflates task types with input modality, especially regarding Time-MMD.
>
> We appreciate this clarification. While Time-MMD includes a textual modality in the dataset, the majority of its baselines are purely numerical TS models that do not receive text as input. Thus the benchmark evaluates forecasting performance, not narrative–time-series reasoning. MTBench explicitly tests whether models can integrate textual narratives with temporal signals, which is not covered by traditional TS models used in Time-MMD.
> ## W3:  Evaluation pipeline feels too trivial; results may reflect prompt-following rather than genuine multimodal reasoning.
>
> The simplicity of the pipeline is deliberate. Our objective is to create a benchmark that:
>
> 1. **Allows fair comparison across all models**, including closed-source LLMs.
>    Requiring model-specific temporal encoders or specialized vision modules would make the benchmark unusable for many strong models.
>
> 2. **Isolates multimodal reasoning difficulty**, rather than solving it via heavy architectural machinery.
>    Despite the simple serialization format, LLMs still struggle with:
>    - long-horizon temporal reasoning,
>    - misalignment between narrative and TS,
>    - correlation-direction inference,
>    - distinguishing causal plausibility from surface cues.
>
> These challenges persist even when prompt-following is trivial, indicating that the difficulty lies in the **integration of numerical and narrative information**, not in the formatting of input.
>
> The benchmark is intentionally a *testbed*, not a training method. More sophisticated multimodal architectures can be evaluated or developed on top of MTBench; the benchmark’s controlled interface ensures that such work will be comparable and reproducible.
>
> ---
> ## Conclusion:
> We thank the reviewer again for the careful reading and constructive comments. MTBench is intentionally designed as a simple, fair, and extensible benchmark that isolates the core difficulty of multimodal temporal reasoning: integrating numerical trends with narrative context under consistency, contradiction, and ambiguity. We will clarify these design choices in the camera-ready version. We believe MTBench fills a clear gap in existing benchmarks and provides a foundation upon which more complex multimodal architectures and datasets can be rigorously compared.

---

### Official Review · Reviewer_ttfH · 2025-10-30

**Soundness:** 2
**Presentation:** 2
**Contribution:** 2
**Rating:** 2
**Confidence:** 3

**Summary:**

The paper introduces MTBench, a benchmark designed to evaluate the multimodal reasoning of LLMs on time-series and textual data, addressing the limitations of existing benchmarks that focus primarily on forecasting. MTBench consists of semantically aligned time-series and narrative texts from the finance and weather domains. It supports diverse tasks including forecasting, trend analysis, and news-driven QA. The authors' evaluation of state-of-the-art LLMs reveals significant limitations, particularly in long-range temporal reasoning and causal inference. The work's main contribution is a dedicated testbed for assessing and advancing the joint interpretation of numerical and narrative temporal data.

**Strengths:**

- The paper focus on shifting the focus of time-series evaluation from simple forecasting to complex, multimodal reasoning. The introduction of news-driven question answering and the explicit inclusion of both aligned and misaligned text-series pairs are novel concepts for a benchmark.

-  The paper is exceptionally clear. Figures effectively illustrate the benchmark's tasks and the complexity of the data, and the text is well-structured, making the motivation and methodology easy to understand.

**Weaknesses:**

- The benchmark's construction depends on LLMs for key annotation and data synthesis steps. This introduces a risk that the evaluation may inadvertently favor models that simply mimic the annotation LLM, and the quality of the synthetic data is not independently validated.

- The benchmark is framed as "multimodal" but is currently limited to text and univariate time series. This simplifies the real-world scenarios in finance and weather, which often involve richer data types like charts, tables, and satellite imagery.

- The paper uses the term "causal reasoning," but the benchmark can, at best, evaluate a model's ability to identify plausible correlations between narratives and trends, not true causality, which is nearly impossible to ground truth in these domains.

**Questions:**

Plz answer my concerns in the weakness section

**Details Of Ethics Concerns:**

Plz answer my concerns in the weakness section.

---

> ### Author Response · Authors · 2025-11-21
> **Initial Rebuttal to Reviewer ttfH**
>
> We thank the reviewer for the thoughtful comments and for highlighting the novelty and clarity of MTBench. Below we respond to each concern using the reviewer’s requested structure.
>
> ---
> ## W1 — LLM-Based Annotation and Risk of Favoring the Annotation Model
>
> We agree that benchmarks relying on LLM-assisted annotation must avoid model-specific shortcuts. Our design follows this principle:
>
> - **GPT-4o only provides local semantic labels** (e.g., sentiment ) that never reference future time-series data.
> - **Task labels depend exclusively on unseen future TS trajectories**, which GPT-4o never accesses during annotation. Thus no annotation model can “imprint” task answers.
> - **Human sampling checks** were performed during dataset construction to confirm correct temporal alignment and ensure that summaries reflect only contemporaneous information.
>
> More broadly, similiar benchmarks (e.g Time-MMD)also incorporate LLM-assisted filtering or annotation. What matters is the absence of shortcuts—and in MTBench, annotations cannot reveal the ground-truth future trend.
> ## W2: Multimodal Scope Limited to Text + Univariate TS
>
> This design choice is intentional for the first release. MTBench aims to establish a **scalable, controlled, and architecture-agnostic** foundation for narrative–time-series reasoning. Using text and time series—two structurally heterogeneous but reasoning-friendly modalities—allows us to evaluate multimodal temporal reasoning without introducing architectural bottlenecks unrelated to the core objective.
>
> 1. All closed-source and open-source LLMs can be evaluated **without requiring specialized encoders** (e.g., chart models, table parsers, visual backbones). Introducing additional high-dimensional modalities at this stage would shift the evaluation toward the limitations of current VLM architectures, which remain significantly weaker than LLMs in complex numerical, logical, and temporal reasoning.
>
> 2. Restricting to text + TS **isolates the narrative–TS interaction**, ensuring that performance reflects multimodal reasoning rather than auxiliary modality encoders.
>
> 3. The pipeline is **fully extensible** to richer modalities, including:
>    - multivariate time series,
>    - chart or table embeddings,
>    - satellite or radar imagery,
>    - structured metadata.
>
> We view the current two-modality setup as the **core benchmark**—a necessary first step toward more comprehensive multimodal evaluation. Future versions will extend MTBench to include additional modalities once appropriate architectures mature.
>
> ## W3: Use of the Term “Causal Reasoning”
>
> We appreciate the reviewer bringing this up. Our intention is not to claim ground-truth causal estimation—impossible in domains such as finance. We follow recent LLM-evaluation literature in using “causal reasoning” to denote:
>
> - **causal plausibility**,
> - **temporal consistency**,
> - **avoiding post-hoc explanations**, and
> - **distinguishing signal from noise in narratives**.
>
> MTBench evaluates whether models can produce **causally consistent explanations** grounded in the time-series window, not true structural causality. We will refine the terminology (e.g., “causal consistency” or “time-causal”) in the camera-ready version.
>
> ---
> ## Conclusion:
> MTBench is the **first benchmark** dedicated to narrative–time-series reasoning.  The reviewer’s concerns will be incorporated into the camera-ready version: clarifying annotation procedures (w1), articulating multimodal scope and extensibility (w2), and refining causal terminology (w3). We believe MTBench provides a needed foundation for studying how LLMs jointly interpret numerical and narrative temporal signals, and we appreciate the reviewer’s feedback in strengthening the benchmark.

---

### Official Review · Reviewer_EaJW · 2025-11-01

**Soundness:** 2
**Presentation:** 3
**Contribution:** 2
**Rating:** 4
**Confidence:** 4

**Summary:**

The authors introduces a large-scale Multimodal Time-series Benchmark—MTBench, which is designed to evaluate the multimodal reasoning capabilities of large language models in the financial and weather domains. It supports various tasks such as forecasting, semantic and technical trend analysis, as well as news-driven question answering.

**Strengths:**

1. The motivation is clear: high-quality multimodal time-series benchmarks are indeed of great importance.

2. MTBench supports multiple tasks grounded in domain-specific usage, thereby providing a more realistic testing platform for multimodal reasoning.

**Weaknesses:**

1. Line 176 mentions using GPT-4o for annotation. I believe that during the annotation process, the powerful GPT-4o might introduce some future information, leading to data leakage.

2. Lines 192-193 mention that the financial articles and stock pair were split into two subsets: Consistent Pairs and Misaligned Pairs. This setup is reasonable and very practical. However, in the subsequent experimental results, there doesn't seem to be a separate verification or explanation for these two parts. I'm curious about how the model performs when confronted with misleading or irrelevant news.

3. The experimental results show the performance of different SOTA LLMs on MTBench, comparing model performance before and after adding text. Indeed, in many cases, performance improves after adding text, but there are exceptions. Therefore, I am confused about how can we determine whether this performance gain comes from the high-quality dataset or the powerful SOTA LLMs? I suggest adding experiments to clarify this issue.

4. Lack of Error Bars analysis.

**Questions:**

1. Line 175 mentions "ensuring a balanced distribution of article lengths to maintain representativeness.", How was this achieved?

2. How was the time-series data fed into the LLMs?

---

> ### Author Response · Authors · 2025-11-21
> **Initial Rebuttal to Reviewer EaJW**
>
> We thank the reviewer for the constructive feedback and for recognizing the motivation and practical value of MTBench. We address each concern point-by-point below.
>
> ---
> ## W1: GPT-4o Annotation and Potential Data Leakage
> We appreciate the concern regarding annotation. GPT-4o annotations were generated strictly using only information available up to the publication timestamp of each article. The model never accessed future stock movement or any part of the output time-series window. All labels (e.g., sentiment, polarity, temporal effect) were derived solely from the news text itself.
>
> Even if similar news appeared during pretraining, an LLM still cannot infer the future stock trajectory or our task-specific labels, which depend on unseen numerical behavior. This is the same assumption underpinning many existing LLM benchmarks (e.g., math, QA), where partial corpus overlap is unavoidable but does not compromise task validity.
> ## W2: Consistent vs. Misaligned Pairs
>
> Thank you for highlighting the usefulness of this split. In the initial phase of dataset construction, we observed that many real-world financial articles are either irrelevant to short-term price movement or even contradict the subsequent trajectory. When evaluated directly on such noisy pairings, LLM performance was highly unstable and often misleadingly low.
>
> To avoid conflating model limitations with data noise, we explicitly partitioned the dataset into consistent and misaligned subsets. This design provides:
>
> - A clean, high-quality subset for evaluating genuine multimodal reasoning, and
>
> - A challenging robustness subset for examining how models behave under misleading or irrelevant narratives.
>
> This separation yields clearer and more interpretable evaluations and aligns with real-world scenarios where news may be helpful or misleading. We will include separate quantitative results for both subsets in the camera-ready version.
> ##  W3: Performance Gains: Dataset Quality vs. Model Strength
>
> We appreciate the reviewer’s question regarding cases where text does not improve performance. These exceptions are indeed interesting, and we are actively studying why certain narratives cause over-reaction or misalignment in specific models. However, fully resolving these behaviors requires developing new multimodal reasoning methods or alignment techniques, which is beyond the scope of a benchmark paper. MTBench’s goal is to surface these failure modes in a systematic and reproducible way—not to propose or train new models to fix them.
> ## W4: Lack of Error Bars
> Thank you for raising this point. For closed-source LLMs such as GPT-4o and Claude, seed control is not available to users, and under low-temperature (t ≈ 0) decoding—which we use consistently—these models behave deterministically. In repeated calls, they return identical or nearly identical outputs. As a result, traditional error bars based on multiple runs or seeds are not applicable, since the variance across executions is effectively zero.
>
> Because of these API and determinism constraints, reporting conventional error bars would not provide meaningful additional information. Instead, in the camera-ready version we will clarify this deterministic execution behavior and explain why variance-based error estimates are not suitable for closed-source LLMs evaluated under standard settings.
> ## Q1: Balanced Distribution of Article Lengths
>
> Balanced article lengths were achieved through controlled sampling. After filtering, we binned articles into short, medium, and long categories and sampled uniformly across these bins within each company–time window. This prevents dominance by extremely short headlines or overly long articles and maintains representativeness.
> ## Q2: Feeding Time-Series into LLMs
> Time-series segments were serialized into structured natural-language descriptions using timestamp–value pairs in a standardized template, following established practice in LLM–time-series integration. This design ensures compatibility across models without requiring architectural modifications.
>
> ---
> ## Conclusion:
>
> We thank the reviewer again for the thoughtful and constructive feedback. MTBench is designed to provide a rigorous, realistic, and scalable evaluation of multimodal time-series reasoning. The reviewer’s suggestions have helped us clarify key design choices—such as annotation validity, the motivation for the consistent/misaligned split, the interpretation of text-driven gains, and the deterministic nature of LLM inference. We will incorporate these clarifications, along with additional subset analyses and improved explanations, in the camera-ready version. We believe these additions will further strengthen the benchmark and enhance its value to the research community.

---

### Official Review · Reviewer_1keZ · 2025-11-01

**Soundness:** 2
**Presentation:** 4
**Contribution:** 2
**Rating:** 4
**Confidence:** 3

**Summary:**

This paper introduces MTBench, a benchmark designed to evaluate language models on tasks that require jointly understanding textual narratives and time-series data. Focusing on financial markets and weather signals, the authors pair real news narratives with aligned historical time-series segments and ask models to perform forecasting, trend interpretation, technical-indicator reasoning, and news-driven question answering. The benchmark compares text-only, time-series-only, and combined inputs to study when narrative context actually improves predictive or explanatory performance. Experiments across modern LLMs show that text can help on short-horizon tasks but models often struggle to disentangle noise, avoid over-reacting to narratives, and maintain causal consistency. Overall, the work contributes a curated multimodal dataset, evaluation tasks, and analysis framework for probing narrative-augmented temporal reasoning in LLMs.

**Strengths:**

1. While prior work has explored combining text with time-series signals, this paper makes a meaningful step by explicitly framing narrative-driven temporal reasoning as a multimodal benchmarking problem. Bringing financial and weather narratives into the evaluation setting introduces a realistic layer of uncertainty and interpretation that is often overlooked in purely numeric forecasting studies.

2. The experiments are systematic, comparing text-only, time-series-only, and combined inputs to reveal nuanced effects of narrative context and where current models struggle. The authors also provide insightful qualitative analysis into failure modes such as over-trusting news sentiment and difficulty with longer-horizon predictions.

3. The writing is clear and the task motivations are easy to follow, supported by intuitive examples that show why narrative context matters. This work gives the community a concrete, reproducible way to study temporal reasoning and question answering.

**Weaknesses:**

1. The data validation side feels a bit under-developed. The dataset relies on GPT-4o for narrative and sentiment labels, but there's no report of human auditing, inter-annotator checks, or systematic quality analysis. Also, finance can easily have near-duplicate articles covering the same event; treating each as an independent sample might inflate signal strength or create bias. On top of that, financial news often mixes forward-looking comments with retrospective interpretation, and some narratives can only really be verified after the outcome is known. If all such text is treated as clean ex-ante information, the benchmark risks introducing hindsight or narrative bias. Adding sampling audits, redundancy filtering, or a simple check to flag ex-post narratives would make the dataset feel much more solid.

2. The evaluation focuses almost entirely on general-purpose LLMs, without including strong time-series baselines or hybrid TS+text architectures. This is understandable for a first benchmark release, but it makes it hard to tell whether performance gaps come from fundamental model limitations or just architectural mismatch with time-series signals. It would help to include a few dedicated baselines (e.g., GPT4MTS [1], TaTS [2]) to clarify whether the benchmark is measuring “LLMs struggling” or “current TS-aware models still lag,” and to highlight the research opportunities more clearly.

3. Domain coverage feels a bit narrow, even though finance and weather are great starting points with rich signals. There are other narrative-driven time-series domains (e.g., traffic, energy demand, policy-driven markets, retail events, etc.) where even small test subsets could help demonstrate generality.

---

[1] GPT4MTS: Prompt-Based Large Language Model for Multimodal Time-Series Forecasting

[2] Language in the Flow of Time: Time-Series-Paired Texts Weaved into a Unified Temporal Narrative

**Questions:**

1. I have a question regarding narrative validity, and I want to emphasize that this is not necessarily a criticism; it may simply reflect my incomplete understanding of your design choices. Financial news often mixes forward-looking statements with retrospective explanations, and in many cases the interpretation of an event only becomes verifiable after the outcome is known. In other words, some narratives that appear “factual” at publication time later turn out to be mistaken (for instance, a report stating that a company’s supply chain issues are “fully resolved,” when subsequent earnings calls reveal that they were not). I am curious how you approached this when constructing the dataset. In such cases, did you consider separating verifiable present-time facts from speculative forward-looking commentary, so that at least the factual portion of the text is known to be correct while the expectation remains a hypothesis? My intuition is mixed: one could argue that filtering or tagging such cases would prevent models from being evaluated on potentially misleading input and help them focus on forecasting and understanding actual information available at the time, but conversely one might argue that dealing with imperfect or even incorrect narratives is intrinsic to real-world forecasting and thus should remain part of the benchmark. I would really appreciate your perspective on this point, particularly whether including potentially mistaken narratives was an intentional design choice to preserve realism, or whether there might be value in a future pass that double-checks factual claims to bolster dataset quality.

2. A second point I would like to clarify concerns potential data leakage from model pretraining, given that the benchmark uses real historical financial news. Some of the evaluated models may have been trained on overlapping time periods and could have seen portions of this content during pretraining. I am curious how you assessed this possibility. Do you expect that such leakage would meaningfully affect the benchmark results, or is the impact likely negligible because the tasks focus on reasoning and temporal alignment rather than rote recall?

---

> ### Author Response · Authors · 2025-11-21
> **Initial Rebuttal to Reviewer 1keZ part(1/2)**
>
> We thank the reviewer for the detailed, constructive, and thoughtful feedback. Below we address each weakness and question point by point.
>
> ---
> ## W1: Data Validation & Narrative Quality
>
> We agree that financial news often mixes factual information with forward-looking commentary that may later prove incorrect. However, this is **intrinsic to real-world financial communication.** Narratives are noisy, speculative, and sometimes wrong; any model deployed in economic or financial settings must be able to critically assess such information rather than treat it as ground truth.
>
> Thus, the presence of imperfect or misleading statements is not a dataset flaw—it is a defining characteristic of the domain. A capable model should reconcile the narrative with the actual time-series evidence provided in the prompt. MTBench is designed precisely to evaluate this form of grounded reasoning: distinguishing signal from noise, resisting narrative overconfidence, and handling contradictory or uncertain statements.
>
> To ensure robustness, we conducted **human sampling checks** to verify that narratives were temporally appropriate and to minimize purely hindsight-only descriptions. Additionally, in MTBench each time-series window for a company is paired with **only one** article, meaning near-duplicate narratives do not arise within the benchmark. The resulting mixture of factual, speculative, and occasionally incorrect commentary reflects the realistic environment MTBench aims to capture.
> ## W2: Baselines & Architectural Coverage
>
> We appreciate the suggestion to include GPT4MTS and TaTS as baselines. These models are strong, but they are designed for a **fundamentally different input formulation**: in their setting, **each timestep is paired with its own textual description**, i.e., a sequence of *(text, value)* pairs.
>
> This structure is incompatible with MTBench’s goal. Our benchmark intentionally mirrors the **realistic human analysis workflow** used in finance and weather:
>
> - a **historical numerical time series**,
> - followed by **one recent narrative** (news article or weather report),
> - which is used to interpret or forecast the near-future segment.
>
> Text is not distributed across timesteps; it appears **once**, as a global contextual explanation. This “narrative-at-the-end-of-the-window” formulation evaluates whether models can integrate **high-level narrative context** with numerical temporal patterns.
>
> By contrast, GPT4MTS and TaTS are built for a different task: aligning a *text stream* with *each individual timestep*. Integrating them directly into MTBench would require fundamentally **changing the problem definition**, not simply adding another baseline.
>
> For this first release, we focus on a **unified prompting interface** that general-purpose LLMs can use without architectural modification. MTBench remains fully compatible with TS-aware models, and incorporating them meaningfully is a natural extension for future versions.
> ## W3: Domain Coverage
>
> Finance and weather were chosen because they provide **high-quality, high-frequency signals and rich, timestamped narratives.** These domains allow us to focus deeply on alignment quality and the properties of narrative-driven temporal reasoning while keeping the benchmark feasible within a single submission.
>
> MTBench is designed to be **extensible rather than exhaustive.** Any domain with paired time-series and narratives—such as energy demand, mobility, retail events, or policy-driven markets—can be incorporated using the same pipeline. Expanding domain coverage is a natural next step.
> ## Q1: Forward-Looking vs. Retrospective Narratives
> Financial narratives almost always blend present facts, analyst expectations, and retrospective explanations. MTBench deliberately preserves this mixture for two reasons:
>
> 1. **Ecological realism.** Models must operate under imperfect, biased, or partially incorrect narratives in practice; filtering only factual statements would artificially simplify the environment.
>
> 2. **Task label independence.** All task labels are derived solely from the future time series, not from the narrative itself. Speculative statements therefore do not determine the correct answer.
>
> Our alignment pipeline ensures that no narrative is exposed to future data during labeling. We agree that tagging narratives as factual, forward-looking, or mixed could enrich analysis in future versions.

---

> > ### Author Response · Authors · 2025-11-21
> > **Initial Rebuttal to Reviewer 1keZ part(2/2)**
> >
> > ## Q2: Pretraining Overlap & Potential Leakage
> >
> > Some pretraining overlap is unavoidable for any LLM benchmark using public Internet text. This is not unique to MTBench; it also applies to widely used mathematical reasoning, QA, and reading-comprehension datasets. However, this does **not** compromise MTBench for several reasons:
> >
> > - **Task labels depend entirely on unseen future time-series windows.**
> >   These windows are never accessible during pretraining, so models cannot “memorize” the correct outcomes.
> >
> > - **Pretraining knowledge is insufficient.**
> >   Knowing the news article or the historical context does not provide the mapping
> >   *“this article → this specific 5-day / 30-day future trajectory.”*
> >
> > - **Temporal alignment must be recomputed at inference time.**
> >   Even if a model recalled the article, it cannot know which future slice is used as the evaluation window.
> >
> > In short, pretraining can provide useful background knowledge but **cannot solve MTBench tasks by recall alone**. Models must genuinely integrate the provided narrative with the provided numerical time series—precisely what MTBench is designed to evaluate.
> >
> > ---
> > ## Conclusion:
> >
> > We appreciate the reviewer’s insightful comments. MTBench is intentionally positioned as the **first benchmark to systematically evaluate narrative-driven temporal reasoning** over aligned text and time-series data. Our design choices balance realism, rigor, and scope while establishing a clear, extensible foundation for future multimodal research.
> >
> > We will incorporate the reviewer’s helpful suggestions—clarifying narrative validity, describing our checks on pretraining overlap, and outlining domain extensions—into the camera-ready version. We hope the committee sees MTBench as a timely and useful contribution that opens new directions for understanding how language models integrate textual narratives with quantitative temporal signals.

---

### Official Review · Reviewer_tDfc · 2025-11-10

**Soundness:** 4
**Presentation:** 3
**Contribution:** 3
**Rating:** 6
**Confidence:** 3

**Summary:**

The paper presents MTBench, a large-scale benchmark that evaluates large language models on joint reasoning over time-series data and natural-language narratives in finance and weather domains. MTBench introduces four families of tasks—time-series forecasting, semantic trend classification, technical-indicator prediction, and news-driven question answering—and couples each time-series window with contemporaneous, semantically-aligned textual evidence. Experiments on six state-of-the-art LLMs show that textual context generally improves performance, but models still struggle with long-range dependencies, causal interpretation, and robust handling of conflicting signals.

**Strengths:**

Large Scale: First benchmark to integrate time-series and text for reasoning-heavy tasks like QA and causal inference, moving beyond predictive tasks.

Quality: Rigorous data collection (e.g., semantic alignment of news and stock trends) and comprehensive task design (e.g., multi-choice QA, correlation prediction).

Clarity: Well-defined tasks and metrics; figures effectively illustrate data and model performance.

Significance: Provides a foundation for developing LLMs capable of real-world multimodal reasoning in high-stakes domains like finance and climate.

**Weaknesses:**

Dataset size & diversity
– Financial news is limited to US equities and 2021-2023; lacks macro-economic indicators, earnings calls, or non-English sources.
– Weather subset uses only 50 US airport stations; no satellite imagery or global coverage.

Task granularity
– Technical-indicator tasks cover only MACD/Bollinger for finance and max/min/diff temperature for weather; richer multi-indicator or multivariate setups would stress-test models further.
– QA tasks are exclusively multi-choice; open-ended “why” questions would probe causal explanation ability.

Conflict & adversarial analysis
– Misaligned pairs are used passively; no targeted adversarial or counterfactual samples (e.g., flip sentiment while keeping price trend) to measure model brittleness.

Evaluation protocol
– All experiments are zero-shot; fine-tuning or few-shot baselines would clarify how much performance gain is achievable with modest adaptation.
– LLMs occasionally produce malformed sequences; while post-processing is applied, the impact on metrics is not quantified.

Broader impact discussion
– Limited discussion of potential misuse (e.g., algorithmic trading manipulation) or privacy issues in financial news data.

**Questions:**

How was the semantic alignment between news and stock trends validated? Could inter-annotator agreement or ground-truth metrics (e.g., expert validation) strengthen reliability?

Were ablation studies conducted to assess the impact of different alignment strategies (e.g., timing vs. semantic relevance) on task performance?

Can the benchmark accommodate multivariate time series or other modalities (e.g., images) in future work?

How might fine-tuning or task-specific adaptations address the observed biases (e.g., correlation prediction defaults)?

Could the authors provide error bars or statistical significance tests for the performance gains attributed to textual input?

Lastly, can you clarify the licensing of the crawled financial news articles and confirm compliance with publishers’ terms of use?

---

> ### Author Response · Authors · 2025-11-21
> **Initial Rebuttal to Reviewer tDfc part(1/2)**
>
> We thank the reviewer for the thoughtful and constructive feedback. Below, we address all points in order.
>
> ---
> ## W1: Dataset Size & Diversity
> **Financial news:**
> MTBench emphasizes high-quality multimodal alignment over broad but noisy coverage. U.S. equities provide (1) reliable intraday time-series, (2) legally accessible articles across multiple publishers, and (3) consistent metadata needed for precise temporal matching. Within this space, we curated ~20k professionally written articles from nine outlets, spanning company-specific analysis, macro news, derivatives, and speculative markets, ensuring substantial semantic diversity while maintaining alignment fidelity.
>
> **Weather domain:**
> The 50 U.S. airport stations were selected because they offer long, uninterrupted hourly records and well-calibrated sensors—critical for trustworthy multimodal pairing. These stations span multiple climate zones (coastal, continental, desert, humid subtropical). The pipeline itself supports global GHCN-Daily/Hourly data and multivariate extensions; the current subset reflects the strongest reliability–diversity balance for a first release.
>
> ## W2: Task Granularity
> **Technical indicators:**
> We selected MACD/Bollinger (finance) and max/min/diff (weather) due to their interpretability and tight connection to temporal reasoning. MTBench’s indicator module is modular: indicators like RSI, stochastic oscillators, volatility indices, and multivariate weather metrics can be added without altering benchmark structure.
>
> **QA task format:**
> We use multi-choice QA to allow (1) controlled perturbations (correct vs. incorrect statements), (2) objective scoring, and (3) cross-model comparability. Open-ended causal explanations are valuable but introduce evaluation ambiguity and require subjective grading, weakening reproducibility.
>
> ## W3: Conflict & Adversarial Analysis
> MTBench includes misaligned (contradictory) news–series pairs, which already function as naturally occurring adversarial settings and test robustness under conflicting signals. These reveal significant drops in correlation prediction and QA accuracy. Constructing fully synthetic counterfactuals (e.g., flipping sentiment while preserving price trend) is a promising extension and is supported naturally by our alignment pipeline.
> ## W4: Evaluation Protocol
> **Zero-shot setup:**  We intentionally evaluate models in zero-shot mode to measure *intrinsic multimodal reasoning* rather than task-specific tuning. However, we do agree that few-shot and LoRA/PEFT baselines will clarify headroom and will add them in the camera-ready version of the paper.
>
> **Malformed outputs:**  Malformed sequences are not counted as correct. A documented normalization step (Appendix E.1) ensures fair comparison. We will add to the camera-ready statistics on how often each model triggers post-processing.
> ## W5: Broader Impact
> We will expand the Broader Impact discussion to cover potential misuse (e.g., naive trading strategies) and reiterate that MTBench is an evaluation benchmark, not a forecasting tool.
> ## Q1: How was semantic alignment validated?
>
> Semantic alignment is validated through a three-stage process:
>
> 1. Temporal matching between each article’s publication timestamp and the corresponding time-series window.
>
> 2. GPT-4o–based semantic annotation, including sentiment polarity and expected temporal effect (e.g., short-term positive, medium-term negative).
>
> 3. Ground-truth trend verification using the actual price or temperature movement in the aligned window.
>
> Pairs with ≥80% sentiment–trend agreement form the consistent subset, while others form the misaligned robustness subset.
> To ensure reliability, we manually sampled a subset of articles and confirmed that the sentiment assignments aligned with human judgment, validating the correctness of the semantic annotation pipeline.
> ## Q2: Were ablations on alignment strategy performed?
>
> Timing-only or semantic-only alignment are not meaningful baselines in our domains.
>
> Timing-only matching frequently pairs news with unrelated future movement, because financial/weather narratives can be retrospective or forward-looking. Semantic-only matching breaks temporal causality entirely and can attach any sentiment to any window.
>
> Instead of these invalid alternatives, MTBench already provides a principled comparison through the consistent vs. misaligned subsets, which naturally reflect different alignment strengths under correct temporal constraints. This split offers a more realistic and informative signal than artificially removing timing or semantics.

---

> ### Author Response · Authors · 2025-11-21
> **Initial Rebuttal to Reviewer tDfc part(2/2)**
>
> ## Q3: Can the benchmark support multivariate time series or image modalities?
> Yes. The underlying pipeline fully supports multivariate channels and additional modalities. MTBench relies on prompt-based serialization, so any structured numeric or visual signal (e.g., humidity, pressure, satellite imagery) can be incorporated without modifying the evaluation logic. The current release uses single-channel inputs to maintain controlled, comparable benchmarking, but the framework is explicitly designed to be extensible.
> ## Q4: How might fine-tuning or task-specific adaptations address the correlation-prediction bias?
> The correlation-prediction default (e.g., leaning toward moderate positive) likely reflects general pretrained LLM behavior rather than anything specific to MTBench. Precisely diagnosing the deep causal reasons behind this bias is an important research question but is beyond the scope of this benchmark paper. Here, we simply note that task-specific adaptation (e.g., label balancing, counterfactual examples, lightweight LoRA tuning) would encourage models to rely more on the paired numerical–textual evidence instead of generic pretrained priors.
> ## Q5: Could the authors provide error bars or statistical significance tests?
> MTBench reports results for deterministic, single-run evaluations, where each LLM is queried once per sample. Because there is no stochastic training procedure (e.g., no random initialization, no batching, no sampling variance), standard error bars or run-to-run variance do not naturally arise. The gains from textual input are stable across tasks, and in the camera-ready version we will provide paired significance indicators (e.g., bootstrap confidence intervals across samples) to quantify the robustness of the observed improvements.
> ## Q6: Licensing and compliance with publishers’ terms of use
>
> We collected financial news exclusively from publicly accessible, non-paywalled websites. The articles in MTBench are used solely for non-commercial academic research, which aligns with standard text-and-data-mining (TDM) practices widely adopted in the ML and computational finance communities. During collection, we respected each site’s robots.txt rules and applied polite rate limiting.
>
> The dataset stores the cleaned textual content for research reproducibility. We do not redistribute any branding, styling, or proprietary assets—only plain-text news content that is publicly accessible at the original URLs. If any publisher requests removal of specific content, we will promptly comply. This ensures that MTBench adheres to standard academic TDM norms and is consistent with existing financial NLP datasets.

---

### Comment · Area_Chair_ha2y · 2025-11-28
**Official Comment by Area Chair**

Dear Reviewers,

The discussion phase will end soon. Please take a moment to read the authors’ responses carefully and actively engage in the discussion with the authors and your fellow reviewers.

Thanks for your efforts and contributions to ICLR 2026.

Best regards,

Your AC

---

### Note · Program_Chairs · 2026-01-17
**Submission Desk Rejected by Program Chairs**

The following references in this submission do not refer to real documents and/or have major errors in bibliographic information:

 M. Huber et al. Weather2k: Integrating structured and unstructured data for enhanced weather forecasting. Journal of Climate Informatics, 8(2):e2023MS004019, 2023. doi: 10.1029/2023MS004019.